# On the Convergence and Sample Efficiency of Variance-Reduced Policy Gradient Method

**Junyu Zhang**
Department of Industrial Systems Engineering and Management
National University of Singapore
Singapore, 119077
junyuz@nus.edu.sg

**Chengzhuo Ni**
Department of Electrical and Computer Engineering
Princeton University
Princeton, NJ, 08544
chengzhuo.ni@princeton.edu

**Zheng Yu**
Department of Electrical and Computer Engineering
Princeton University
Princeton, NJ, 08544
zhengy@princeton.edu

**Csaba Szepesvari**
Department of Computer Science
University of Alberta
Edmonton, Alberta, Canada T6G 2E8
szepesva@ualberta.ca

**Mengdi Wang**
Department of Electrical and Computer Engineering
Princeton University
Princeton, NJ, 08544
mengdiw@princeton.edu

## Abstract

Policy gradient (PG) gives rise to a rich class of reinforcement learning (RL) methods. Recently, there has been an emerging trend to accelerate the existing PG methods such as REINFORCE by the *variance reduction* techniques. However, all existing variance-reduced PG methods heavily rely on an uncheckable importance weight assumption made for every single iteration of the algorithms. In this paper, a simple gradient truncation mechanism is proposed to address this issue. Moreover, we design a Truncated Stochastic Incremental Variance-Reduced Policy Gradient (TSIVR-PG) method, which is able to maximize not only a cumulative sum of rewards but also a general utility function over a policy's long-term visiting distribution. We show an $\tilde{\mathcal{O}}(\epsilon^{-3})$ sample complexity for TSIVR-PG to find an $\epsilon$-stationary policy. By assuming the overparameterization of policy and exploiting the hidden convexity of the problem, we further show that TSIVR-PG converges to global $\epsilon$-optimal policy with $\tilde{\mathcal{O}}(\epsilon^{-2})$ samples.

## 1 Introduction

In this paper, we investigate the theoretical properties of Policy Gradient (PG) methods for Reinforcement Learning (RL) [43]. In view of RL as a policy optimization problem, the PG method

35th Conference on Neural Information Processing Systems (NeurIPS 2021).

parameterizes the policy function and conduct gradient ascent search to improve the policy. In this paper, we consider the soft-max policy parameterization

$$\pi_\theta(a|s) = \frac{\exp\{\psi(s,a;\theta)\}}{\sum_{a'}\exp\{\psi(s,a';\theta)\}} \tag{1}$$

where $(s,a)$ is a state-action pair and $\psi$ is some smooth function. Potentially, one can set the function $\psi$ to be some deep neural network with weights $\theta$ and input $(s,a)$. The main problem considered in this paper is the policy optimization for a *general utility* function:

$$\max_\theta R(\pi_\theta) := F(\lambda^{\pi_\theta}), \tag{2}$$

where $F$ is a general smooth function, and $\lambda^{\pi_\theta}$ denotes the unnormalized state-action occupancy measure (also referred to as the visitation measure). For any policy $\pi$ and initial state distribution $\xi$,

$$\lambda^\pi(s,a) := \sum_{t=0}^{+\infty} \gamma^t \cdot \mathbb{P}\Big(s_t = s, a_t = a \,\big|\, \pi, s_0 \sim \xi\Big), \tag{3}$$

where $\gamma$ stands for the discount factor and $\mathbb{P}$ denotes the probability of a certain event. When $F$ is linear, the problem reduces to the standard policy optimization problem where the objective is to maximize a cumulative sum of rewards. When $F$ is nonlinear, problem (2) goes beyond standard Markov decision problems: examples include the max-entropy exploration [16], risk-sensitive RL [50], certain set constrained RL [27], and so on.

In the standard cumulative-return case (i.e., $F$ is linear), numerous works have studied PG methods in various scenarios, see e.g. [47, 5, 58, 22, 21, 37, 38, 23]. When directly optimizing over the policy space without any parameterization, the policy mirror descent (PMD) method [23] achieves an $\tilde{\mathcal{O}}(\epsilon^{-2})$ sample complexity to find an $\mathcal{O}(\epsilon)$-optimal solution. However, for the more practical parameterized policy optimization, to the authors' best knowledge, the most recent variant of PG methods, using the SARAH/Spider stochastic variance reduction technique [13, 29], find a local $\epsilon$-stationary policy using $\tilde{\mathcal{O}}(\epsilon^{-3})$ samples [48, 33]. This poses a contrast with the known $\tilde{O}(\epsilon^{-2})$ sample complexity results that can be achieved by various value-based methods [4, 42, 41] and are provably matching information-theoretic lower bounds [12, 3, 4]. In this paper, we attempt to close this gap and prove an $\tilde{\mathcal{O}}(\epsilon^{-2})$ sample complexity bound for a PG method. Most importantly, when it comes to PG estimation, the application of the variance reduction technique typically relies on certain off-policy PG estimator, resulting in the difficulty of distribution shift. We notice that none of the existing variance-reduced PG methods attempt to address this challenge. Instead, they directly make an uncheckable assumption that the variance of the importance weight is bounded for every policy pair encountered in running the algorithm, see e.g. [31, 48, 49, 33]. In this paper, we propose a simple gradient truncation mechanism to fix this issue.

Next, let us go beyond cumulative return and consider policy optimization for a general utility where $F$ may be nonlinear. However, much less is known in this setting. The nonlinearity of $F$ invalidates the concept of Q-function and value function, leading to the failure of policy gradient theorem [44]. To overcome such difficulty, [51] showed that the policy gradient for the general utilities is the solution to a min-max problem. However, estimating a single PG is highly nontrivial in this case. It is still unclear how to make PG methods to use samples in a most efficient way.

In this paper, we aim to investigate the convergence and sample efficiency of the PG method, using episodic sampling, for both linear $F$ (i.e., cumulative rewards) and nonlinear $F$ (i.e., general utility). Observe that problem (2) is an instance of the Stochastic Composite Optimization (SCO) problem [45, 46]: $\min_x f(\mathbb{E}_\nu[g_\nu(x)])$, which involves an inner expectation that corresponds to the occupancy measure $\lambda^\pi$. Motivated by this view point, we attempt to develop stochastic policy gradient method with provable finite-sample efficiency bounds.

**Main results.** Our main results are summarized below.

- We propose the TSIVR-PG algorithm to solve problem (2) via episodic sampling. It provides a conceptually simple stochastic gradient approach for solving general utility RL.

- We provide a gradient truncation mechanism to address the distribution shift difficulty in variance-reduced PG methods. Such difficulty has never been addressed in previous works.

- We show that TSIVR-PG finds an $\epsilon$-stationary policy using $\tilde{O}(\epsilon^{-3})$ samples if $F$ and $\psi$ are general smooth functions. When $F$ is concave and $\psi$ satisfies certain overparameterization condition, we show that TSIVR-PG obtains a gloal $\epsilon$-optimal policy using $\tilde{O}(\epsilon^{-2})$ samples.

**Technical contribution.** Our analysis technique is also of independent interest in the relating areas.

- For stochastic composite optimization (SCO), most existing algorithms require estimating the Jacobian matrix of the inner mapping, which corresponds to $\nabla_\theta \lambda^{\pi_\theta}$ in our setting. This is in practice prohibitive if the Jacobian matrix has high dimensions, which is exactly the case in our problem. Unlike SCO algorithms such as [24, 54, 53, etc.], our analysis enables us to avoid the Jacobian matrix estimation.

- For the stochastic variance-reduced gradient methods, our analysis implies a convergence of SARAH/Spider methods to global optimality and a new $\mathcal{O}(\epsilon^{-2})$ sample complexity for nonconvex problems with "hidden convexity" structure, which has not been studied in the optimization community yet.

## 2 Related Works

Policy gradient gives rises to a rich family of RL algorithms, such as REINFORCE and many of its variants [47, 5, 58], as well as extensions such as the natural policy gradient methods [20, 32], the actor-critic methods [22, 21, 28], the trust-region policy optimization [37, 57, 39], and the proximal policy optimization method [38, 25], etc. In this paper we mainly focus on REINFORCE-type methods, where many of them need $\tilde{\mathcal{O}}(\epsilon^{-4})$ samples to find an $\epsilon$-stationary solution, including the vanilla REINFORCE [47], as well as its variants with baseline [58, 43] and GPOMDP [5], etc. By incorporating the stochastic variance reduction techniques, the sample efficiency of PG methods can be further improved. In [31], the SVRG [19] variance reduction scheme is adopted and an $\mathcal{O}(\epsilon^{-4})$ sample complexity is achieved, which is later improved to $\mathcal{O}(\epsilon^{-10/3})$ by [48]. With additional Hessian information, [40] achieved an $\mathcal{O}(\epsilon^{-3})$ complexity. By utilizing a more efficient SARAH/Spider [29, 13] variance reduction scheme, people are able to achieve $\mathcal{O}(\epsilon^{-3})$ sample complexity without second-order information [48, 33]. We would like to comment that these results are only for finding $\epsilon$-stationary (rather than near-optimal) solutions, and all of them requires an uncheckable condition on the importance weights in every iteration.

Recently, for cumulative reward, a series of works have started to study the convergence of policy gradient method to global optimal solutions [1, 14, 56, 6, 26, 7, 9, 55]. In particular, [51] exploited the hidden convexity property of the MDP problem and established the convergence to global optimality for general utility RL problem, as long as the policy gradient can be computed exactly.

Our approach is related to the stochastic composite optimization (SCO) [45, 46]. For the general composition problem, there have been numerous developments, including momentum-based and multi-time-scale algorithms [45, 46, 15], and various composite stochastic variance-reduced algorithms [24, 17, 52, 54]. Our approach is also inspired by variance reduction techniques that were initially used for stochastic convex optimization, see [19, 36, 11, 29]; and were later on extended to the stochastic nonconvex optimization problems [2, 34, 18, 35, 13, 30]. In particular, we will utilize the SARAH/Spider scheme [13, 30].

## 3 Problem Formulation

Consider an MDP with a general utility function, denoted as $\mathrm{MDP}(\mathcal{S}, \mathcal{A}, \mathcal{P}, \gamma, F)$, where $\mathcal{S}$ is a finite state space, $\mathcal{A}$ is a finite action space, $\gamma \in (0, 1)$ is a discount factor, and $F$ is some general utility function. For each state $s \in \mathcal{S}$, a transition to state $s' \in \mathcal{S}$ occurs when selecting an action $a \in \mathcal{A}$ following the distribution $\mathcal{P}(\cdot|a, s)$. For each state $s \in \mathcal{S}$, a policy $\pi$ gives a distribution $\pi(\cdot|s)$ over the action space $\mathcal{A}$. Let $\xi$ be the initial state distribution and let the unnormalized state-action occupancy measure $\lambda^\pi$ be defined by (3), we define the general utility function $F$ as a smooth function of the occupancy measure, and the goal of the general utility MDP is to maximize $F(\lambda^\pi)$. With the policy $\pi_\theta$ being parameterized by (1), we propose to solve problem (2), which is

$$\max_\theta R(\pi_\theta) := F\left(\lambda^{\pi_\theta}\right).$$

For notational convenience, we often write $\lambda(\theta)$ instead of $\lambda^{\pi_\theta}$. Such utility function is very general and includes many important problems in RL. We provide a few examples where $F$ are *concave*.

**Example 3.1** (Cumulative reward). *When $F(\lambda^{\pi_\theta}) = \langle r, \lambda^{\pi_\theta} \rangle$, for some $r \in \mathbb{R}^{|\mathcal{S}||\mathcal{A}|}$. Then we recover the standard cumulative sum of rewards:* $R(\pi_\theta) = \mathbb{E}\left[ \sum_{t=0}^{+\infty} \gamma^t \cdot r(s_t, a_t) \,\middle|\, \pi_\theta, s_0 \sim \xi \right]$.

**Example 3.2** (Maximal entropy exploration). *Let $\mu^{\pi_\theta}(s) = (1-\gamma) \sum_a \lambda^{\pi_\theta}(s, a)$, $\forall s \in \mathcal{S}$ be the state occupancy measure, which is the margin of $\lambda^{\pi_\theta}$ over $\mathcal{S}$. Let $F(\cdot)$ be the entropy function, then we recover the objective for maximal entropy exploration [16]:* $R(\pi_\theta) = - \sum_{s \in \mathcal{S}} \mu^{\pi_\theta}(s) \log \mu^{\pi_\theta}(s)$.

**Example 3.3** (RL with Set Constraint). *Let $\mathbf{z}(s_t, a_t) \in \mathbb{R}^d$ be a vector feedback received in each step. The cumulative feedback is $\mathbf{u}(\pi_\theta) := \mathbb{E}\left[ \sum_{t=0}^{+\infty} \gamma^t \cdot \mathbf{z}(s_t, a_t) | s_0 \sim \xi, \pi_\theta \right] = M\lambda^{\pi_\theta}$ for some matrix $M \in \mathbb{R}^{d \times |\mathcal{S}||\mathcal{A}|}$. [27] proposed a set-constrained RL problem which aims to find a policy $\pi$ s.t. $u(\pi) \in U$ for some convex set $U$. This problem can be formulated as an instance of (2) by letting $F(\cdot)$ be the negative squared distance:* $R(\pi_\theta) = -\min_{\mathbf{u}' \in U} \|\mathbf{u}' - \mathbf{u}(\pi_\theta)\|^2$.

# 4 The TSIVR-PG Algorithm

In this section, we propose a Truncated Stochastic Incremental Variance-Reduced Policy Gradient (TSIVR-PG) method, which is inspired by techniques of variance reduction and off-policy estimation. A gradient truncation mechanism is proposed to provably control the importance weights in off-policy sampling.

## 4.1 Off-Policy PG Estimation

**Policy Gradient** First, let us derive the policy gradient of the general utility. Let $V^{\pi_\theta}(r)$ be the cumulative reward under policy $\pi_\theta$, initial distribution $\xi$ and reward function $r$. By Example 3.1, $V^{\pi_\theta}(r) = \langle \lambda(\theta), r \rangle$, the chain rule and policy gradient theorem [44] indicates that

$$\nabla_\theta V^{\pi_\theta}(r) = \left[ \nabla_\theta \lambda(\theta) \right]^\top r = \mathbb{E}_{\xi, \pi_\theta} \left[ \sum_{t=0}^{+\infty} \gamma^t \cdot r(s_t, a_t) \cdot \left( \sum_{t'=0}^{t} \nabla_\theta \log \pi_\theta(a_{t'}|s_{t'}) \right) \right], \qquad (4)$$

where $\nabla_\theta \lambda(\theta)$ is the Jacobian matrix of the vector mapping $\lambda(\theta)$. That is, policy gradient theorem actually provides a way for computing the Jacobian-vector product for the occupancy measure. Following the above observation and the chain rule, we have

$$\nabla_\theta R(\pi_\theta) = [\nabla_\theta \lambda(\theta)]^\top \nabla_\lambda F(\lambda(\theta)) = \nabla_\theta V^{\pi_\theta}(r)|_{r = \nabla_\lambda F(\lambda(\theta))}.$$

Therefore, we can estimate the policy gradient using the typical REINFORCE as long as we pick the "quasi-reward function" as $r := \nabla_\lambda F(\lambda(\theta))$. To find this quasi-reward, we need to estimate the state-action occupancy measure $\lambda(\theta)$ (unless $F$ is linear).

**Importance Sampling Weight** Let $\tau = \{s_0, a_0, s_1, a_1, \cdots, s_{H-1}, a_{H-1}\}$ be a length-$H$ trajectory generated under the initial distribution $\xi$ and the behavioral policy $\pi_{\theta_1}$. For any target policy $\pi_{\theta_2}$, we define the importance sampling weight as

$$\omega_t(\tau|\theta_1, \theta_2) = \frac{\Pi_{h=0}^t \pi_{\theta_2}(a_h|s_h)}{\Pi_{t=0}^h \pi_{\theta_1}(a_h|s_h)}, \qquad 0 \le t \le H - 1. \qquad (5)$$

It is worth noting that such importance sampling weight is inevitable in the stochastic variance reduced policy gradient methods, see [31, 48, 49, 33, 25]. In these works, the authors usually directly assume $\mathrm{Var}(\omega_{H-1}(\tau|\theta_1, \theta_2)) \le W$ for all the policy pairs encountered in every iteration of their algorithms. However, such assumption is too strong and is uncheckable.

Based on the above notation of behavioral and target policies, as long as the importance sampling weights, we present the following off-policy occupancy and policy gradient estimators.

**Off-Policy Occupancy Measure Estimator** Denote $\mathbf{e}_{sa}$ the vector with $(s, a)$-th entry being 1 while other entries being 0. We define the following estimator for $\lambda(\theta_2)$

$$\widehat{\lambda}_\omega(\tau|\theta_1, \theta_2) := \sum_{t=0}^{H-1} \gamma^t \cdot \omega_t(\tau|\theta_1, \theta_2) \cdot \mathbf{e}_{s_t a_t}. \qquad (6)$$

When $\theta_2 = \theta_1$, $\omega_t(\tau|\theta_1, \theta_2) \equiv 1$ and $\widehat{\lambda}_\omega(\tau|\theta_1, \theta_2)$ becomes the on-policy (discounted) empirical distribution, for which we use the simplified notion $\widehat{\lambda}(\tau|\theta_2) := \widehat{\lambda}_\omega(\tau|\theta_2, \theta_2)$.

**Off-Policy Policy Gradient Estimator**  Let $r \in \mathbb{R}^{|\mathcal{S}||\mathcal{A}|}$ be any quasi-reward vector. We aim to estimate the Jacobian-vector product $[\nabla_\theta \lambda(\theta_2)]^\top r$ for target policy $\pi_{\theta_2}$ by

$$\widehat{g}_\omega(\tau|\theta_1, \theta_2, r) := \sum_{t=0}^{H-1} \gamma^t \cdot \omega_t(\tau|\theta_1, \theta_2) \cdot r(s_t, a_t) \cdot \left( \sum_{t'=0}^{t} \nabla_\theta \log \pi_{\theta_2}(a_{t'}|s_{t'}) \right). \tag{7}$$

When $\theta_2 = \theta_1$, $\omega_t(\tau|\theta_1, \theta_2) \equiv 1$ and $\widehat{g}_\omega(\tau|\theta_1, \theta_2, r)$ becomes the on-policy REINFORCE estimator with quasi-reward function $r$. In this case, we use the simplified notion $\widehat{g}(\tau|\theta_2, r) := \widehat{g}_\omega(\tau|\theta_2, \theta_2, r)$.

Estimators $\widehat{\lambda}_\omega(\tau|\theta_1, \theta_2)$ and $\widehat{g}_\omega(\tau|\theta_1, \theta_2, r)$ are almost unbiased. In details, $\|\mathbb{E}_{\tau \sim \pi_{\theta_1}}[\widehat{\lambda}_\omega(\tau|\theta_1, \theta_2)] - \lambda(\theta_2)\| \leq \mathcal{O}(\gamma^H)$ and $\|\mathbb{E}_{\tau \sim \pi_{\theta_1}}[\widehat{g}_\omega(\tau|\theta_1, \theta_2, r)] - [\nabla_\theta \lambda(\theta_2)]^\top r\| \leq \mathcal{O}(H \cdot \gamma^H)$; see details in Appendix E. Therefore the bias due to truncation is almost negligible if $H$ is properly selected.

### 4.2 The TSIVR-PG Algorithm

To achieve the $\tilde{\mathcal{O}}(\epsilon^{-2})$ sample complexity, we propose an epoch-wise algorithm called Truncated Stochastic Incremental Variance-Reduced PG (TSIVR-PG) Algorithm. Let $\theta_0^i$ be the starting point of the $i$-th epoch, TSIVR-PG constructs the estimators for $\lambda(\theta_0^i)$, quasi-reward $\nabla_\lambda F(\lambda(\theta_0^i))$ and the policy gradient $\nabla_\theta F(\lambda(\theta_0^i))$ by

$$\lambda_0^i = \frac{1}{N} \sum_{\tau \in \mathcal{N}_i} \widehat{\lambda}(\tau|\theta_0^i), \quad r_0^i = \nabla_\lambda F(\lambda_0^i) \quad \text{and} \quad g_0^i = \frac{1}{N} \sum_{\tau \in \mathcal{N}_i} \widehat{g}(\tau|\theta_0^i, r_0^i). \tag{8}$$

where $\mathcal{N}_i$ is a set of $N$ independent length-$H$ trajectories sampled under $\pi_{\theta_0^i}$. When $j \geq 1$,

$$\lambda_j^i = \frac{1}{B} \sum_{\tau \in \mathcal{B}_j^i} \left( \widehat{\lambda}(\tau|\theta_j^i) - \widehat{\lambda}_\omega(\tau|\theta_j^i, \theta_{j-1}^i) \right) + \lambda_{j-1}^i, \qquad r_j^i = \nabla_\lambda F(\lambda_j^i) \tag{9}$$

$$g_j^i = \frac{1}{B} \sum_{\tau \in \mathcal{B}_j^i} \left( \widehat{g}\left(\tau|\theta_j^i, r_{j-1}^i\right) - \widehat{g}_\omega\left(\tau|\theta_j^i, \theta_{j-1}^i, r_{j-2}^i\right) \right) + g_{j-1}^i, \tag{10}$$

where $\mathcal{B}_j^i$ is a set of $B$ independent length-$H$ trajectories sampled under $\pi_{\theta_j^i}$, and we default $r_{-1}^i := r_0^i$. Specifically, $\widehat{g}_\omega(\tau|\theta_j^i, \theta_{j-1}^i, r_{j-2}^i)$ is used instead of $\widehat{g}_\omega(\tau|\theta_j^i, \theta_{j-1}^i, r_{j-1}^i)$ for independence issue. The details of the TSIVR-PG algorithm are stated in Algorithm 1.

---

**Algorithm 1:** The TSIVR-PG Algorithm

---

1 **Input:** Initial point $\theta_0^1 = \tilde{\theta}_0$; batch sizes $N$ and $B$; sample trajectory length $H$; stepsize $\eta$; epoch length $m$; gradient truncation radius $\delta$.

2 **for** *Epoch* $i = 1, 2, ...,$ **do**

3     **for** *Iteration* $j = 0, ..., m-1$ **do**

4         **if** $j == 0$ **then**

5             Sample $N$ trajectories under policy $\pi_{\theta_0^i}$ of length $H$, collected as $\mathcal{N}_i$.

6             Compute estimators $\lambda_0^i$, $r_0^i$ and $g_0^i$ by (8). Default $r_{-1}^i := r_0^i$.

7         **else**

8             Sample $B$ trajectories under policy $\pi_{\theta_j^i}$ with length $H$, collected as $\mathcal{B}_j^i$.

9             Compute estimators $\lambda_j^i$, $r_j^i$ and $g_j^i$ by (9) and (10).

10         Update the policy parameter by a truncated gradient ascent step:

$$\theta_{j+1}^i = \begin{cases} \theta_j^i + \eta \cdot g_j^i & , \text{ if } \eta\|g_j^i\| \leq \delta, \\ \theta_j^i + \delta \cdot g_j^i / \|g_j^i\| & , \text{ otherwise.} \end{cases} \tag{11}$$

11     Set $\theta_0^{i+1} = \tilde{\theta}_i = \theta_m^i$.

---

It is worth noting that the truncated gradient step (11) is equivalent to a trust region subproblem:

$$\theta_{j+1}^i = \underset{\|\theta - \theta_j^i\| \leq \delta}{\operatorname{argmax}} F(\lambda(\theta_j^i)) + \langle g_j^i, \theta - \theta_j^i \rangle + \frac{1}{2\eta}\|\theta - \theta_j^i\|^2 \tag{12}$$

where the approximate Hessian matrix is simply chosen as $(\eta)^{-1} \cdot I$.

# 5 Sample Efficiency of TSIVR-PG

In this section, we analyze the finite-sample performance of TSIVR-PG. We first show that TSIVR-PG finds an $\epsilon$-stationary solution with $\tilde{\mathcal{O}}(\epsilon^{-3})$ samples. Given additional assumptions, we show that TSIVR-PG finds a global $\epsilon$-optimal solution with $\tilde{\mathcal{O}}(\epsilon^{-2})$ samples .

## 5.1 Convergence Towards Stationary Points

Since we focus on the soft-max policy parameterization where $\pi_\theta(a|s) = \frac{\exp\{\psi(s,a;\theta)\}}{\sum_{a'} \exp\{\psi(s,a';\theta)\}}$, we make the following assumptions on the parameterization function $\psi$ and the utility $F$.

**Assumption 5.1.** $\psi(s, a; \cdot)$ *is twice differentiable for all $s$ and $a$. There $\exists \ell_\psi, L_\psi > 0$ s.t.*

$$\max_{s\in\mathcal{S},a\in\mathcal{A}} \sup_\theta \|\nabla_\theta \psi(s, a; \theta)\| \leq \ell_\psi \quad and \quad \max_{s\in\mathcal{S},a\in\mathcal{A}} \sup_\theta \|\nabla_\theta^2 \psi(s, a; \theta)\| \leq L_h, \tag{13}$$

*where $\|\cdot\|$ stands for $L_2$ norm and spectral norm for vector and matrix respectively.*

**Assumption 5.2.** *$F$ is a smooth and possibly nonconvex function. There exists $\ell_{\lambda,\infty} > 0$ such that $\|\nabla_\lambda F(\lambda)\|_\infty \leq \ell_{\lambda,\infty}$. And there exist constants $L_{\lambda,\infty}, L_\lambda > 0$ s.t. it holds for $\forall \lambda, \lambda'$ that*

$$\|\nabla_\lambda F(\lambda) - \nabla_\lambda F(\lambda')\|_\infty \leq L_\lambda \|\lambda - \lambda'\|_2 \quad and \quad \|\nabla_\lambda F(\lambda) - \nabla_\lambda F(\lambda')\|_\infty \leq L_{\lambda,\infty}\|\lambda - \lambda'\|_1.$$

Based on the above assumptions, we have the following supporting lemmas.

**Lemma 5.3.** *Given Assumption 5.1 and 5.2, the following results hold:*
**(i).** *For any policy parameter $\theta$ and any state-action pair $(s, a)$, the following inequalities hold:*
$\|\nabla_\theta \log \pi_\theta(a|s)\| \leq 2\ell_\psi$, $\|\nabla_\theta^2 \log \pi_\theta(a|s)\| \leq 2(L_\psi + \ell_\psi^2)$, *and* $\|\nabla_\theta F(\lambda(\theta))\| \leq \frac{2\ell_\psi \cdot \ell_{\lambda,\infty}}{(1-\gamma)^2}$.
**(ii).** *For any policy parameters $\theta_1$ and $\theta_2$, it holds that $\|\lambda^{\pi_{\theta_1}} - \lambda^{\pi_{\theta_2}}\|_1 \leq \frac{2\ell_\psi}{(1-\gamma)^2} \cdot \|\theta_1 - \theta_2\|$.*
**(iii).** *The objective function $F \circ \lambda(\cdot)$ is $L_\theta$-smooth, with $L_\theta = \frac{4L_{\lambda,\infty}\cdot\ell_\psi^2}{(1-\gamma)^4} + \frac{8\ell_\psi^2 \cdot \ell_{\lambda,\infty}}{(1-\gamma)^3} + \frac{2\ell_{\lambda,\infty}\cdot(L_\psi+\ell_\psi^2)}{(1-\gamma)^2}$.*

To measure the convergence, we propose to use the gradient mapping defined as follows:

$$\mathcal{G}_\eta(\theta) = \frac{\theta_+ - \theta}{\eta}, \quad \text{where} \quad \theta_+ = \begin{cases} \theta + \eta \cdot g & \text{, if } \eta\|g\| \leq \delta, \\ \theta + \delta \cdot g/\|g\| & \text{, otherwise} \end{cases}$$

where $g = \nabla_\theta F(\lambda(\theta))$. We remark that, $\mathbb{E}[\|\mathcal{G}_\eta(\theta_j^i)\|^2]$ is more suitable for the ascent analysis of the truncated gradient updates, compared with the commonly used $\mathbb{E}[\|\nabla_\theta F(\lambda(\theta_j^i))\|^2]$. Note that $\mathcal{G}_\eta(\theta) = \nabla F(\lambda(\theta))$ if $\|\mathcal{G}_\eta(\theta)\| \leq \delta$ and $\|\nabla F(\lambda(\theta))\|$ is bounded for any $\theta$. Based on such observation, we have the following lemma to validate the choice of the proposed stationarity measure.

**Lemma 5.4.** *For any random vector $\theta$, $\mathbb{E}[\|\mathcal{G}_\eta(\theta)\|] \leq \epsilon$ implies $\mathbb{E}[\|\nabla_\theta F(\lambda(\theta))\|] \leq \mathcal{O}(\delta^{-1} \cdot \epsilon)$.*

Based on the notion of $\mathcal{G}_\eta$, we characterize the per-iteration ascent as follows.

**Lemma 5.5.** *Let the iterates be generated by Algorithm 1. Then it holds that*

$$F(\lambda(\theta_{j+1}^i)) \geq F(\lambda(\theta_j^i)) + \frac{\eta}{4}\|\mathcal{G}_\eta(\theta_j^i)\|^2 + \left(\frac{1}{2\eta} - L_\theta\right)\|\theta_{j+1}^i - \theta_j^i\|^2 - \left(\frac{\eta}{2} + \frac{1}{2L_\theta}\right)\|\nabla_\theta F(\lambda(\theta_j^i)) - g_j^i\|^2.$$

This suggests us to bound mean-squared-error $\mathbb{E}[\|\nabla_\theta F(\lambda(\theta_j^i)) - g_j^i\|^2]$. For this purpose, we need to bound the importance sampling weight, by utilizing the soft-max form of policy parameterization (1).

**Lemma 5.6.** *For any behavioral policy $\pi_{\theta_1}$ and target policy $\pi_{\theta_2}$ parameterized by (1), the importance weight satisfies $\omega_t(\tau|\theta_1, \theta_2) \leq \exp\{2(t+1)\ell_\psi\|\theta_1 - \theta_2\|\}$, for $\forall 0 \leq t \leq H - 1$.*

Since TSIVR-PG only uses importance weights for two consecutive iterations $\theta_j^i, \theta_{j-1}^i$ while forcing $\|\theta_j^i - \theta_{j-1}^i\| \leq \delta$ by the truncated gradient step (11), we have $\omega_{H-1}(\tau|\theta_j^i, \theta_{j-1}^i) \leq \exp\{2H\ell_\psi \delta\}$ w.p. 1. As we will see later, the effective horizon $H$ only has a mild magnitude of $\mathcal{O}((1-\gamma)^{-1} \cdot \log(1/\epsilon))$, the truncation radius only need to satisfy $\delta = \mathcal{O}(H^{-1}\ell_\psi^{-1})$ s.t. $\omega_{t-1}(\tau|\theta_j^i, \theta_{j-1}^i) = \mathcal{O}(1)$, for $\forall t \leq H-1$. Consequently, combining Lemma 5.3, 5.6 and Lemma B.1 of [48] gives the following result.

**Lemma 5.7.** *Let policy $\pi_\theta$ be parameterized by (1) with function $\psi$ satisfying Assumption 5.1. Suppose behavioral policy $\pi_{\theta_1}$ and target policy $\pi_{\theta_2}$ satisfy $\|\theta_1 - \theta_2\| \leq \delta$, then*

$$\mathbb{E}[\omega_t(\tau|\theta_1, \theta_2)] = 1 \quad and \quad \mathrm{Var}\left(\omega_t(\tau|\theta_1, \theta_2)\right) \leq C_\omega(t+1) \cdot \|\theta_1 - \theta_2\|^2,$$

*where $\tau$ is sampled under policy $\pi_{\theta_1}$, and $C_\omega(t) = t\left(4\ell_\psi^2(t + \frac{1}{2}) + 2L_\psi\right)(e^{4\ell_\psi \delta t} + 1)$.*

As a result, we can bound the mean-squared-error of the $g_j^i$ as follows.

**Lemma 5.8.** *For the PG estimators $g_j^i$, we have*

$$\mathbb{E}\left[\|g_j^i - \nabla_\theta F(\lambda(\theta_j^i))\|^2\right] \leq \frac{C_1}{N} + C_2\gamma^{2H} + \frac{C_3}{B} \cdot \sum_{j'=1}^{j} \mathbb{E}\left[\|\theta_{j'-1}^i - \theta_{j'}^i\|^2\right] + C_4\mathbb{E}\left[\|\theta_{j-1}^i - \theta_j^i\|^2\right]$$

*for some constants $C_1, .., C_4 > 0$. In case $j = 0$, we default $\sum_{j'=1}^{0} \cdot = 0$.*

The expression of constants $C_i$'s are complicated, we provide their detailed formula in the appendix. If we set $H = \mathcal{O}\left(\frac{\log(1/\epsilon)}{1-\gamma}\right)$ and $\delta \leq \frac{1}{2H\ell_\psi}$, then $C_i$ only depends polynomially on the Lipschitz constants, $\log(\epsilon^{-1})$, and $(1-\gamma)^{-1}$. Combining Lemma 5.5, 5.8, and 5.4 gives Theorem 5.9.

**Theorem 5.9.** *For Algorithm 1, we choose $H = \frac{2\log(1/\epsilon)}{1-\gamma}$, $\delta = \frac{1}{2H\ell_\psi}$, $B = m = \epsilon^{-1}$, $N = \epsilon^{-2}$, $\eta = \frac{1}{1+(C_3+C_4)/L_\theta^2} \cdot \frac{1}{2L_\theta}$. After running the algorithm for $T = \epsilon^{-1}$ epochs and output $\theta_{out}$ from $\{\theta_j^i\}_{j=0,\cdots,m-1}^{i=1,\cdots,T}$ uniformly at random, we have $\mathbb{E}[\|\mathcal{G}_\eta(\theta_{out})\|] \leq \mathcal{O}(\epsilon)$. The total number of samples is $T \times ((m-1)B + N) \times H = \tilde{\mathcal{O}}(\epsilon^{-3})$. By Lemma 5.4, we also have $\mathbb{E}[\|\nabla_\theta F(\lambda(\theta_{out}))\|] \leq \mathcal{O}(\epsilon)$.*

### 5.2 Convergence Towards Global Optimality

Next, we provide a mechanism to establish the convergence of TSIVR-PG to global optimality. For this purpose, we introduce the hidden convexity of the general utility RL problem. In addition to the smoothness of $F$ (Assumption 5.2), we further assume its concavity, formally stated as follows.

**Assumption 5.10.** *Function $F$ is a concave function of the state-action occupancy measure.*

Let $\mathcal{L}$ be the image of the mapping $\lambda(\theta)$. Then the parameterized *policy optimization* problem (2) can be rewritten as an equivalent *occupancy optimization* problem:

$$\max_\theta F(\lambda(\theta)) \quad \Longleftrightarrow \quad \max_{\mu \in \mathcal{L}} F(\mu). \tag{14}$$

When the policy parameterization is powerful enough to represent any policy, the image $\mathcal{L}$ is a convex polytope, see e.g. [10]. Since $F$ is concave, the occupancy optimization problem is a *convex optimization* problem. In this case, if the mapping $\lambda(\cdot)$ is invertible (see [51]), we may view the original problem (2) as a reformulation of a convex problem by a change of variable: $\theta = \lambda^{-1}(\mu)$. We call this property "hidden convexity". However, requiring $\lambda(\cdot)$ to be invertible is too restrictive, and it doesn't even hold for simple soft-max policy with $\psi(s, a; \theta) = \theta_{sa}$ where multiple $\theta$ correspond to a same policy. Therefore, we adopt a weaker assumption where (i). $\pi_\theta$ can represent any policy (ii). a continuous inverse $\lambda^{-1}(\cdot)$ can be locally defined over a subset of $\theta$.

**Assumption 5.11.** *For policy parameterization of form (1), $\theta$ overparametrizes the set of policies in the following sense. (i). For any $\theta$ and $\lambda(\theta)$, there exist (relative) neighbourhoods $\theta \in \mathcal{U}_\theta \subset B(\theta, \delta)$ and $\lambda(\theta) \in \mathcal{V}_{\lambda(\theta)} \subset \lambda(B(\theta, \delta))$ s.t. $\left(\lambda|_{\mathcal{U}_\theta}\right)(\cdot)$ forms a bijection between $\mathcal{U}_\theta$ and $\mathcal{V}_{\lambda(\theta)}$, where $\left(\lambda|_{\mathcal{U}_\theta}\right)(\cdot)$ is the confinement of $\lambda$ onto $\mathcal{U}_\theta$. We assume $\left(\lambda|_{\mathcal{U}_\theta}\right)^{-1}(\cdot)$ is $\ell_\theta$-Lipschitz continuous for any $\theta$. (ii). Let $\pi_{\theta^*}$ be the optimal policy. Assume there exists $\bar{\epsilon}$ small enough, s.t. $(1-\epsilon)\lambda(\theta) + \epsilon\lambda(\theta^*) \in \mathcal{V}_{\lambda(\theta)}$ for $\forall \epsilon \leq \bar{\epsilon}, \forall \theta$.*

Based on Assumption 5.11, we replace Lemma 5.5 with the following lemma.

**Lemma 5.12.** *For $\forall \epsilon < \bar{\epsilon}$ with $\bar{\epsilon}$ defined in Assumption 5.11, it holds for all iterations that*

$$F(\lambda(\theta^*)) - F(\lambda(\theta_{j+1}^i)) \leq (1 - \epsilon)\left(F(\lambda(\theta^*)) - F(\lambda(\theta_j^i))\right) \tag{15}$$
$$+ \left(L_\theta + \frac{1}{2\eta}\right)\frac{2\epsilon^2 \ell_\theta^2}{(1-\gamma)^2} - \left(\frac{1}{2\eta} - L_\theta\right)\|\theta_{j+1}^i - \theta_j^i\|^2 + \frac{1}{L_\theta}\|g_j^i - \nabla_\theta F(\lambda(\theta_j^i))\|^2.$$

The analysis of Lemma 5.12 is very different from its nonconvex optimization counterpart (Lemma 5.5). Next, we derive the sample complexity of the TSIVR-PG algorithm given Lemma 5.12 and 5.8.

**Theorem 5.13.** *For TSIVR-PG method (Algorithm 1), let $\epsilon \in (0, \bar{\epsilon})$ be the target accuracy. If we choose $H, m, B, N$ and $\delta$ according to Theorem 5.9. and we let the stepsize to be small enough s.t. $\eta \leq \frac{1}{2L_\theta + 8(C_3 + C_4)/L_\theta}$, then after at most $T = \log_2(\epsilon^{-1})$ epochs, $\mathbb{E}\big[F(\lambda(\theta^*)) - F(\lambda(\tilde{\theta}_T))\big] \leq \mathcal{O}(\epsilon)$. The total number of samples taken is $T \times ((m-1)B + N) \times H = \tilde{\mathcal{O}}(\epsilon^{-2})$.*

## 6 Numerical Experiments

### 6.1 Maximizing Cumulative Reward.

In this experiment, we aim to evaluate the performance of the TSIVR-PG algorithm for maximizing the cumulative sum of reward. As the benchmarks, we also implement the SVRPG [49], the SRVR-PG [48], the HSPGA [33], and the REINFORCE [47] algorithms. Our experiment is performed on benchmark RL environments including the FrozenLake, Acrobot and Cartpole that are available from OpenAI gym [8], which is a well-known toolkit for developing and comparing reinforcement learning algorithms. For all the algorithms, their batch sizes are chosen according to their theory. In details, let $\epsilon$ be any target accuracy. For both TSIVR-PG and SRVR-PG, we set $N = \Theta(\epsilon^{-2})$, $B = m = \Theta(\epsilon^{-1})$. For SVRPG, we set $N = \Theta(\epsilon^{-2})$, $B = \Theta(\epsilon^{-4/3})$ and $m = \Theta(\epsilon^{-2/3})$. For HSPGA, we set $B = \Theta(\epsilon^{-1})$, other parameters are calculated according to formulas in [33] given $B$. For REINFORCE, we set the batchsize to be $N = \Theta(\epsilon^{-2})$. The parameter $\varepsilon$ and the stepsize/learning rate are tuned for each individual algorithm using a grid search. For each algorithm, we run the experiment for multiple times with random initialization of the policy parameters. The curve is obtained by first calculating the moving average of the most recent 50 episodes, and then calculate the median of the return over the outcomes of different runs. The upper and lower bounds of the shaded area are calculated as the $\frac{1}{4}$ and $\frac{3}{4}$ quantiles over the outcomes. We run the experiment for 10 times for the FrozenLake environment and 50 times for the other environments. The detailed parameters used in the experiments are presented in the Appendix.

**FrozenLake** The FrozenLake8x8 environment is a tabular MDP with finite state and action spaces. For this environment, the policy is parameterized with $\psi(s, a; \theta) = \theta_{sa}$.

**Cartpole and Acrobot** Both the Cartpole environment and the Acrobot environment are environments with a discrete action space and a continuous state space. For both environments, we use a neural network with two hidden layers with width 64 for both layers to model the policy.

**Result** We plot our experiment outcomes in Figure 6.1. The experiments show that given enough episodes, all of the algorithms are able to solve the tasks, achieving nearly optimal returns. And as expected, the REINFORCE algorithm takes the longest time to find the optimal policy. While the other algorithms yield a faster convergence speed, the TSIVR-PG algorithm consistently outperforms the other benchmark algorithms under all of the environments, showing the advantage of our method.

### 6.2 Validating the $\tilde{\mathcal{O}}(\epsilon^{-2})$ Sample Complexity

Besides the comparison between different benchmark algorithms, we also perform a validation experiment showing that for certain environments, the convergence rate of TSIVR-PG is close to the theoretical guarantee. Because the parameters $N, B, m$ are dependent on the target accuracy $\epsilon$, in this section we adopt a different way to set up these parameters: We first set a fixed epoch $E$, and perform experiments using different values of the parameter $N$. The parameter $B$ and $m$ are set according to our choice of $N$ by $B = m = \sqrt{N}$. The performance of the algorithm output is calculated as the average score of the last few episodes, which is then averaged over 10 independent runs. Again, we use the FrozenLake8x8 environment to do the experiment. Because FrozenLake8x8 is a tabular

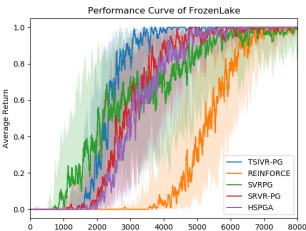 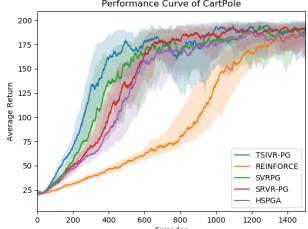 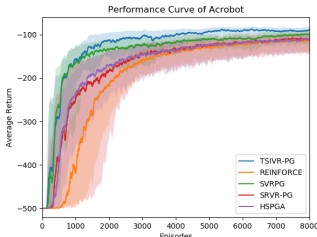

Figure 1: The performance curves of TSIVR-PG and benchmark algorithms under different environments. The curve is the median return over multiple runs and the shaded areas are calculated as the $\frac{1}{4}$ and $\frac{3}{4}$ quantiles of the experiment outcomes.

environment whose transition and reward function can be easily obtained from the document, we can calculate it's optimal value simply by value iteration, which takes $0.4146$ when we choose $\gamma = 0.99$. In this way, we calculate the gap between the algorithm return and the optimal value, and get log-log figure w.r.t. the gap and the number of episodes calculated by $E(N + Bm) = 2EN$.

**Result** The result is shown in the first sub-figure of Figure 6.3, where the blue curve is the gap between the average return of experiment outcome and the optimal value and the shaded area is the range of one standard deviation of the logarithm value. In addition, we add a orange dotted line to fit the convergence curve, whose slope takes value $-0.496$, which nearly matches the $O(\epsilon^{-2})$ theoretical bound (slope $-0.5$).

### 6.3 Maximizing Non-linear Objective Function

The TSIVR-PG algorithm is designed not only to solve typical RL problems, but is also able to solve a broader class of problems where the objective function is a general concave function. Unfortunately, none of the benchmark algorithms proposed in the previous section have the ability to solve this kind of problem. To evaluate the performance of our algorithm, we choose another benchmark algorithm, which is the MaxEnt algorithm [16]. In the experiment, we use FrozenLake8x8 environment since it's more tractable to compute $\lambda$ for a discrete state space. We set the objective function as

$$F(\lambda) = \sum_{s \in \mathcal{S}} \log \left( \sum_{a \in \mathcal{A}} \lambda_{s,a} + \sigma \right),$$

where $\sigma$ is a fixed small constant. We choose $\sigma = 0.125$ in our experiment. The orders of $N, B, m$ are set in the same way as those in section 6.1. For the MaxEnt algorithm, note that in the original paper, the nonlinear objective function assumes the input value is the stationary state distribution $d^\pi$, but the input value can easily be changed into our $\lambda$ without changing the steps of the algorithm much. The result is illustrated in Fig. 6.3. From the result, we may see that our algorithm consistently outperforms the benchmark.

## 7 Broader Impact

There has been an emerging trend of applying stochastic variance reduction technique to enhance the performance of the policy gradient methods. However, all the existing variance-reduced policy gradient methods depend on an algorithm-dependent assumption that is made to every single iteration in running the algorithm, which actually may not be satisfied by many of these algorithms. We propose a simple yet effective mechanism to fix such dilemma for applying the SARAH/Spider variance reduction scheme in policy gradient methods. Our analysis can also be applied to other schemes such as STORM and Hybrid SARAH-SGD. Beyond that, we also show how the hidden convexity and overparameterization of the RL problem can help stochastic variance-reduced policy gradient methods to converge to global optima and yield better sample complexity.

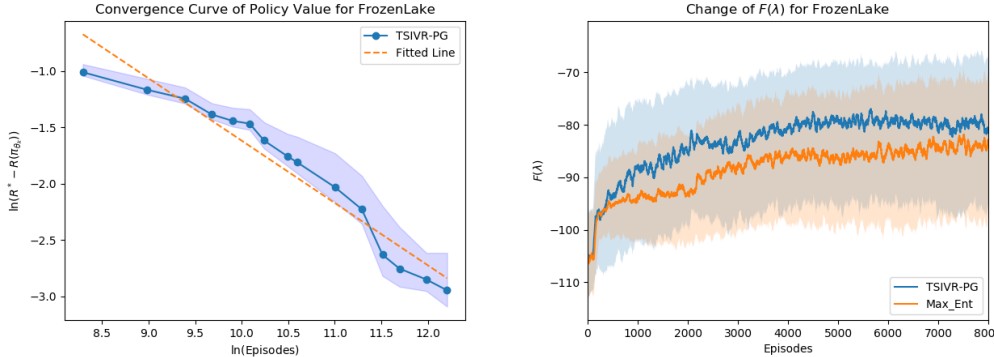

Figure 2: Left: Empirical Evaluation of the Convergence Rate of TSIVR-PG. The optimality gap achieved by TSIVR-PG decreases as the sample size increases, nearly matching the $\epsilon^{-2}$ sample complexity theory (orange line). Right: Performance Curve ofTSIVR-PG and MaxEnt for Maximizing Non-linear Objective Functions. The curve is the median return over 10 runs and the shaded areas are calculated as the $\frac{1}{4}$ and $\frac{3}{4}$ quantiles of the experiment outcomes.

## 8   Limitation

To compute the policy gradient for the general utility function, one needs to compute gradient of the function F with respect to the state-action occupancy measure. Although for many instances, including standard cumulative sum of reward (Example 3.1) and set constrained RL (Example 3.3), estimating the occupancy measure is not necessary. There are many instances where estimating the occupancy measure is unavoidable. Due to the curse of dimensionality, the need for estimating the occupancy measure is a potential limitation if the state and action spaces are continuous and high-dimensional. One potential solution is to incorporate a function approximation for the occupancy measure, in a similar style of the Q-function approximation in the actor-critic method. We leave this for future development.

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
