# A Proof of Lemma 5.3

*Proof.* Let us prove the arguments of this lemma one by one.

**Proof of (i).** Note that the policy $\pi_\theta$ is parameterized by $\pi_\theta(a|s) = \frac{\exp\{\psi(s,a;\theta)\}}{\sum_{a'}\exp\{\psi(s,a';\theta)\}}$. By direct computation, we have

$$\nabla_\theta \log \pi_\theta(a|s) = \nabla_\theta \psi(s,a;\theta) - \sum_{a'} \pi_\theta(a'|s) \cdot \nabla_\theta \psi(s,a';\theta)$$

$$\nabla_\theta^2 \log \pi_\theta(a|s) = \nabla_\theta^2 \psi(s,a;\theta) + \sum_{a',a''} \pi_\theta(a'|s)\pi_\theta(a''|s) \cdot \nabla_\theta \psi(s,a';\theta)\nabla_\theta \psi(s,a'';\theta)^\top$$

$$- \sum_{a'} \pi_\theta(a'|s) \cdot \left( \nabla_\theta^2 \psi(s,a';\theta) + \nabla_\theta \psi(s,a';\theta)\nabla_\theta \psi(s,a';\theta)^\top \right).$$

Because $\|\nabla_\theta \psi(s,a;\theta)\| \le \ell_\psi$ and $\|\nabla_\theta^2 \psi(s,a;\theta)\| \le L_\psi$ for $\forall s,a,\theta$, we have

$$\|\nabla_\theta \log \pi_\theta(a|s)\| \le \ell_\psi + \sum_{a'} \pi_\theta(a'|s)\ell_\psi = 2\ell_\psi,$$

$$\|\nabla_\theta^2 \log \pi_\theta(a|s)\| \le L_\psi + \sum_{a',a''} \pi_\theta(a'|s)\pi_\theta(a''|s) \cdot \ell_\psi^2 + \sum_{a'} \pi_\theta(a'|s) \cdot \left( L_\psi + \ell_\psi^2 \right) = 2(\ell_\psi^2 + L_\psi).$$

Next, for $\|\nabla_\theta F(\lambda(\theta))\|$, by the chain rule and policy gradient (4), it holds that

$$
\begin{aligned}
\|\nabla_\theta F(\lambda(\theta))\| &= \left\| \mathbb{E}\Big[ \sum_{t=0}^{+\infty} \gamma^t \cdot \frac{\partial F(\lambda(\theta))}{\partial \lambda_{s_t a_t}} \cdot \Big( \sum_{t'=0}^{t} \nabla_\theta \log \pi_\theta(a_{t'}|s_{t'}) \Big) \Big] \right\| \\
&\le \mathbb{E}\Big[ \sum_{t=0}^{+\infty} \gamma^t \cdot \|\nabla_\lambda F(\lambda(\theta))\|_\infty \cdot \Big\| \Big( \sum_{t'=0}^{t} \nabla_\theta \log \pi_\theta(a_{t'}|s_{t'}) \Big) \Big\| \Big] \\
&\le \sum_{t=0}^{+\infty} \gamma^t \cdot 2(t+1)\ell_\psi \ell_{\lambda,\infty} \\
&\le \frac{2\ell_\psi \cdot \ell_{\lambda,\infty}}{(1-\gamma)^2}.
\end{aligned}
\tag{16}
$$

**Proof of (ii).** Define $d(\theta,\theta') := \|\lambda(\theta) - \lambda(\theta')\|_1$. Then

$$\nabla_\theta d(\theta,\theta') = \sum_{s,a} \text{sign}(\lambda^{\pi_\theta}(s,a) - \lambda^{\pi_{\theta'}}(s,a)) \cdot \nabla_\theta \lambda^{\pi_\theta}(s,a),$$

where $\text{sign}(x) := 1$ if $x \ge 0$ and $\text{sign}(x) := -1$ if $x < 0$. Let $\mathbf{e}_{sa}$ be the vector with the $(s,a)$-th entry equal to 1 while other entries equal to 0. Then $\lambda^{\pi_\theta}(s,a) = \langle \lambda^{\pi_\theta}, \mathbf{e}_{sa} \rangle$ equals the cumulative sum of rewards with reward function being $\mathbf{e}_{sa}$. By Policy Gradient Theorem [44], we have

$$
\begin{aligned}
\|\nabla_\theta d(\theta,\theta')\| &= \Big\| \sum_{s,a} \text{sign}(\lambda^{\pi_\theta}(s,a) - \lambda^{\pi_{\theta'}}(s,a)) \cdot \nabla_\theta \lambda^{\pi_\theta}(s,a) \Big\| \\
&\le \sum_{s,a} \|\nabla_\theta \lambda^{\pi_\theta}(s,a)\| \\
&\overset{(a)}{=} \sum_{s,a} \Big\| \mathbb{E}\Big[ \sum_{t=0}^{+\infty} \gamma^t \cdot \mathbf{e}_{sa}(s_t,a_t) \cdot \Big( \sum_{t'=0}^{t} \nabla_\theta \log \pi_\theta(a_{t'}|s_{t'}) \Big) \Big] \Big\| \\
&\overset{(b)}{\le} \sum_{s,a} \mathbb{E}\Big[ \sum_{t=0}^{+\infty} \gamma^t \cdot \mathbf{e}_{sa}(s_t,a_t) \cdot \Big\| \sum_{t'=0}^{t} \nabla_\theta \log \pi_\theta(a_{t'}|s_{t'}) \Big\| \Big] \\
&\overset{(c)}{\le} 2\ell_\psi \cdot \mathbb{E}\Big[ \sum_{t=0}^{+\infty} \gamma^t \cdot (t+1) \cdot \sum_{s,a} \mathbf{e}_{sa}(s_t,a_t) \Big] \\
&\overset{(d)}{=} 2\ell_\psi \cdot \sum_{t=0}^{+\infty} \gamma^t \cdot (t+1) \\
&= \frac{2\ell_\psi}{(1-\gamma)^2}.
\end{aligned}
$$

In the above arguments, (a) is due to (4). (b) is because $\|\mathbb{E}[X]\| \leq \mathbb{E}[\|X\|]$ for any random vector $X$. (c) is due to (i) of Lemma 5.3. (d) is because $\sum_{sa} \mathbf{e}_{sa}(s', a') \equiv 1$. As a result,

$$d(\theta, \theta') \leq d(\theta', \theta') + \frac{2\ell_\psi}{(1-\gamma)^2} \|\theta - \theta'\| = \frac{2\ell_\psi}{(1-\gamma)^2} \|\theta - \theta'\|.$$

This completes the proof of (ii) of Lemma 5.3.

**Proof of (iii).** By the chain rule, $\nabla_\theta F(\lambda(\theta)) = [\nabla_\theta \lambda(\theta)]^\top \nabla_\lambda F(\lambda(\theta))$. Therefore,

$$\|\nabla_\theta F(\lambda(\theta_1)) - \nabla_\theta F(\lambda(\theta_2))\| \tag{17}$$

$$= \|[\nabla_\theta \lambda(\theta_1)]^\top \nabla_\lambda F(\lambda(\theta_1)) - [\nabla_\theta \lambda(\theta_2)]^\top \nabla_\lambda F(\lambda(\theta_2))\|$$

$$\leq \underbrace{\|[\nabla_\theta \lambda(\theta_1)]^\top (\nabla_\lambda F(\lambda(\theta_1)) - \nabla_\lambda F(\lambda(\theta_2)))\|}_{T_1} + \underbrace{\|[\nabla_\theta \lambda(\theta_1) - \nabla_\theta \lambda(\theta_2)]^\top \nabla_\lambda F(\lambda(\theta_2))\|}_{T_2}.$$

For the term $T_1$, by (4), we have

$$T_1 = \|[\nabla_\theta \lambda(\theta_1)]^\top (\nabla_\lambda F(\lambda(\theta_1)) - \nabla_\lambda F(\lambda(\theta_2)))\| \tag{18}$$

$$= \left\|\mathbb{E}\left[\sum_{t=0}^{+\infty} \gamma^t \cdot \left(\frac{\partial F(\lambda(\theta_1))}{\partial \lambda_{s_t a_t}} - \frac{\partial F(\lambda(\theta_2))}{\partial \lambda_{s_t a_t}}\right) \cdot \left(\sum_{t'=0}^{t} \nabla_\theta \log \pi_\theta(a_{t'}|s_{t'})\right)\right]\right\|$$

$$\leq \mathbb{E}\left[\sum_{t=0}^{+\infty} \gamma^t \cdot \|\nabla_\lambda F(\lambda(\theta_1)) - \nabla_\lambda F(\lambda(\theta_2))\|_\infty \cdot \left\|\left(\sum_{t'=0}^{t} \nabla_\theta \log \pi_\theta(a_{t'}|s_{t'})\right)\right\|\right]$$

$$\overset{(a)}{\leq} \sum_{t=0}^{+\infty} \gamma^t \cdot 2(t+1)\ell_\psi \cdot L_{\lambda,\infty} \|\lambda(\theta_1) - \lambda(\theta_2)\|_1$$

$$\overset{(b)}{\leq} \frac{4\ell_\psi^2 \cdot L_{\lambda,\infty}}{(1-\gamma)^4} \cdot \|\theta_1 - \theta_2\|.$$

In the above arguments, (a) is due to Assumption 5.2 and (b) is because $\sum_{t=0}^{+\infty} \gamma^t \cdot (t+1) = \frac{1}{(1-\gamma)^2}$. For the term $T_2$, denote $r = \nabla_\lambda F(\lambda(\theta_2))$, $Q^{\pi_\theta}(s,a)$ be the Q-function for the discounted MDP with reward function $r$. We also define $\mu_{sa}^{\pi_\theta}$ as the occupancy measure with initial state distribution $p(\cdot|s,a)$:

$$\mu_{sa}^{\pi_\theta}(s', a') := \sum_{t=0}^{+\infty} \mathbb{P}\left(s_t = s', a_t = a' \mid \pi_\theta, s_0 \sim p(\cdot|s,a)\right).$$

Note that (ii) of Lemma 5.3 does not rely on the initial state distribution, therefore, $\|\mu_{sa}^{\pi_\theta} - \mu_{sa}^{\pi_{\theta'}}\|_1 \leq \frac{2\ell_\psi}{(1-\gamma)^2} \cdot \|\theta_1 - \theta_2\|$ still holds for any $s, a$. Therefore, by Policy Gradient Theorem (4) and its equivalent form provided in [44], we have

$$T_2 = \|[\nabla_\theta \lambda(\theta_1) - \nabla_\theta \lambda(\theta_2)]^\top \nabla_\lambda F(\lambda(\theta_2))\| \tag{19}$$

$$\overset{(a)}{=} \left\|\sum_{sa} \lambda^{\pi_{\theta_1}}(s,a) Q^{\pi_{\theta_1}}(s,a) \nabla_\theta \log \pi_{\theta_1}(a|s) - \sum_{sa} \lambda^{\pi_{\theta_2}}(s,a) Q^{\pi_{\theta_2}}(s,a) \nabla_\theta \log \pi_{\theta_2}(a|s)\right\|$$

$$\leq \sum_{sa} |\lambda^{\pi_{\theta_1}}(s,a) - \lambda^{\pi_{\theta_2}}(s,a)| \cdot |Q^{\pi_{\theta_1}}(s,a)| \cdot \|\nabla_\theta \log \pi_{\theta_1}(a|s)\|$$

$$+ \sum_{sa} \lambda^{\pi_{\theta_2}}(s,a) \cdot |Q^{\pi_{\theta_1}}(s,a) - Q^{\pi_{\theta_2}}(s,a)| \cdot \|\nabla_\theta \log \pi_{\theta_1}(a|s)\|$$

$$+ \sum_{sa} \lambda^{\pi_{\theta_2}}(s,a) \cdot |Q^{\pi_{\theta_2}}(s,a)| \cdot \|\nabla_\theta \log \pi_{\theta_1}(a|s) - \nabla_\theta \log \pi_{\theta_2}(a|s)\|$$

$$\overset{(b)}{\leq} \frac{2\ell_\psi \cdot \ell_{\lambda,\infty}}{1-\gamma} \|\lambda^{\pi_{\theta_1}} - \lambda^{\pi_{\theta_2}}\|_1 + \frac{2\ell_\psi}{1-\gamma} \max_{sa} |Q^{\pi_{\theta_1}}(s,a) - Q^{\pi_{\theta_2}}(s,a)|$$

$$+ \frac{\ell_{\lambda,\infty}}{(1-\gamma)^2} \max_{sa} \|\nabla_\theta \log \pi_{\theta_1}(a|s) - \nabla_\theta \log \pi_{\theta_2}(a|s)\|$$

In the above argument, the $Q^{\pi_\theta}$ in (a) denotes the Q-function of a discounted MDP with reward function $r$ and policy $\pi_\theta$. (b) is because $\|\nabla_\theta \log \pi_\theta(a|s)\| \leq 2\ell_\psi$, $\sum_{sa} \lambda^{\pi_\theta}(s,a) = (1-\gamma)^{-1}$ and

$Q^{\pi_\theta}(s,a) = \frac{\|\nabla_\lambda F(\lambda(\theta))\|_\infty}{1-\gamma} = \frac{\ell_{\lambda,\infty}}{1-\gamma}$. Therefore, we have

$$T_2 \overset{(a)}{\leq} \frac{4\ell_{\lambda,\infty} \cdot \ell_\psi^2}{(1-\gamma)^3} \cdot \|\theta_1 - \theta_2\| + \frac{2\ell_\psi}{1-\gamma} \max_{sa} |\langle r, \mu_{sa}^{\pi_{\theta_1}} - \mu_{sa}^{\pi_{\theta_2}}\rangle| + \frac{2\ell_{\lambda,\infty}(L_\psi + \ell_\psi^2)}{(1-\gamma)^2} \|\theta_1 - \theta_2\|$$

$$\leq \frac{4\ell_{\lambda,\infty} \cdot \ell_\psi^2}{(1-\gamma)^3} \cdot \|\theta_1 - \theta_2\| + \frac{2\ell_\psi \cdot \|r\|_\infty}{1-\gamma} \|\mu_{sa}^{\pi_{\theta_1}} - \mu_{sa}^{\pi_{\theta_2}}\|_1 + \frac{2\ell_{\lambda,\infty}(L_\psi + \ell_\psi^2)}{(1-\gamma)^2} \|\theta_1 - \theta_2\|$$

$$\overset{(b)}{\leq} \left(\frac{8\ell_{\lambda,\infty} \cdot \ell_\psi^2}{(1-\gamma)^3} + \frac{2\ell_{\lambda,\infty} \cdot (L_\psi + \ell_\psi^2)}{(1-\gamma)^2}\right) \cdot \|\theta_1 - \theta_2\|,$$

where (a) is because $Q^{\pi_\theta}(s,a) = r(s,a) + \gamma \cdot \langle r, \mu_{sa}^{\pi_\theta}\rangle$ and (i) & (ii) of Lemma 5.3; (b) is due to applying (ii) of Lemma 5.3 to $\mu_{sa}^{\pi_\theta}$. Now combining (17), (18) and (19) proves (iii) of Lemma 5.3. $\qquad\square$

## B  Proof of Lemma 5.4

*Proof.* For the ease of notation, let us define $\mathcal{E}$ as the event when $\eta\|\nabla_\theta F(\lambda(\theta))\| < \delta$, and denote $\mathcal{E}^c$ as the complement of the event $\mathcal{E}$. By definition of $\mathcal{G}_\eta(\theta)$, we have

$$\begin{aligned}
\mathbb{E}\big[\|\mathcal{G}_\eta(\theta)\|\big] &= \mathbb{P}(\mathcal{E}) \cdot \mathbb{E}\Big[\|\mathcal{G}_\eta(\theta)\| \,\Big|\, \mathcal{E}\Big] + \mathbb{P}(\mathcal{E}^c) \cdot \mathbb{E}\Big[\|\mathcal{G}_\eta(\theta)\| \,\Big|\, \mathcal{E}^c\Big] \\
&= \mathbb{P}(\mathcal{E}) \cdot \mathbb{E}\Big[\|\nabla_\theta F(\lambda(\theta))\| \,\Big|\, \mathcal{E}\Big] + \mathbb{P}(\mathcal{E}^c) \cdot \frac{\delta}{\eta} \\
&\leq \epsilon
\end{aligned}$$

This indicates that

$$\mathbb{P}(\mathcal{E}) \cdot \mathbb{E}\Big[\|\nabla_\theta F(\lambda(\theta))\| \,\Big|\, \mathcal{E}\Big] \leq \epsilon \quad \text{and} \quad \mathbb{P}(\mathcal{E}^c) \leq \frac{\eta\epsilon}{\delta}.$$

Note that Lemma 5.3 indicates that $\|\nabla_\theta F(\lambda(\theta))\| \leq \frac{2\ell_\psi \cdot \ell_{\lambda,\infty}}{(1-\gamma)^2}$, combined with the above inequalities yields

$$\begin{aligned}
&\mathbb{E}[\|\nabla_\theta F(\lambda(\theta))\|] \\
&= \mathbb{P}(\mathcal{E}) \cdot \mathbb{E}\Big[\|\nabla_\theta F(\lambda(\theta))\| \,\Big|\, \mathcal{E}\Big] + \mathbb{P}(\mathcal{E}^c) \cdot \mathbb{E}\Big[\|\nabla_\theta F(\lambda(\theta))\| \,\Big|\, \mathcal{E}^c\Big] \\
&\leq \epsilon + \frac{\eta\epsilon}{\delta} \cdot \frac{2\ell_\psi \cdot \ell_{\lambda,\infty}}{(1-\gamma)^2} \\
&= \left(1 + \frac{\eta}{\delta} \cdot \frac{2\ell_\psi \cdot \ell_{\lambda,\infty}}{(1-\gamma)^2}\right) \cdot \epsilon.
\end{aligned}$$

This completes the proof. $\qquad\square$

## C  Proof of Lemma 5.5

To prove this lemma, let us first provide a supporting lemma.

**Lemma C.1.** *For Algorithm 1 and any iterates $\theta_j^i$ and $\theta_{j+1}^i$, it holds that*

$$\|\mathcal{G}_\eta(\theta_j^i)\|^2 \leq 2\eta^{-2} \cdot \|\theta_{j+1}^i - \theta_j^i\|^2 + 2 \cdot \|g_j^i - \nabla_\theta F(\lambda(\theta_j^i))\|^2.$$

*Proof.* For Algorithm 1, the truncated gradient update of the iterates can also be written as a gradient projection step: $\theta_{j+1}^i = \mathbf{Proj}_{B(\theta_j^i,\delta)} \left(\theta_j^i + \eta \cdot g_j^i\right)$. Denote

$$\hat{\theta}_{j+1}^i = \mathbf{Proj}_{B(\theta_j^i,\delta)} \left(\theta_j^i + \eta \cdot \nabla_\theta F(\lambda(\theta_j^i))\right).$$

Then by Cauchy's inequality and the non-expansiveness of the projection operator yields

$$\begin{aligned}
&\|\mathcal{G}_\eta(\theta_j^i)\|^2 \\
&= \eta^{-2} \cdot \|\hat{\theta}_{j+1}^i - \theta_j^i\|^2 \\
&\leq 2\eta^{-2} \cdot \|\theta_{j+1}^i - \theta_j^i\|^2 + 2\eta^{-2} \cdot \|\hat{\theta}_{j+1}^i - \theta_{j+1}^i\|^2 \\
&= 2\eta^{-2} \cdot \|\theta_{j+1}^i - \theta_j^i\|^2 + 2\eta^{-2} \cdot \|\mathbf{Proj}_{B(\theta_j^i,\delta)} \left(\theta_j^i + \eta \cdot g_j^i\right) - \mathbf{Proj}_{B(\theta_j^i,\delta)} \left(\theta_j^i + \eta \cdot \nabla_\theta F(\lambda(\theta_j^i))\right)\|^2 \\
&\leq 2\eta^{-2} \cdot \|\theta_{j+1}^i - \theta_j^i\|^2 + 2 \cdot \|g_j^i - \nabla_\theta F(\lambda(\theta_j^i))\|^2.
\end{aligned}$$

$$\square$$

Now we are ready to provide the proof of Lemma 5.5.

*Proof.* By the $L_\theta$-smoothness of the objective function, we have

$$F(\lambda(\theta_{j+1}^i))$$

$$\geq \quad F(\lambda(\theta_j^i)) + \langle \nabla_\theta F(\lambda(\theta_j^i)), \theta_{j+1}^i - \theta_j^i \rangle - \frac{L_\theta}{2}\|\theta_{j+1}^i - \theta_j^i\|^2$$

$$= \quad F(\lambda(\theta_j^i)) + \langle g_j^i, \theta_{j+1}^i - \theta_j^i \rangle - \frac{1}{2\eta}\|\theta_{j+1}^i - \theta_j^i\|^2 + \langle \nabla_\theta F(\lambda(\theta_j^i)) - g_j^i, \theta_{j+1}^i - \theta_j^i \rangle$$

$$\quad + \Big(\frac{1}{2\eta} - \frac{L_\theta}{2}\Big)\|\theta_{j+1}^i - \theta_j^i\|^2$$

$$\overset{(i)}{\geq} \quad F(\lambda(\theta_j^i)) + \langle \nabla_\theta F(\lambda(\theta_j^i)) - g_j^i, \theta_{j+1}^i - \theta_j^i \rangle + \Big(\frac{1}{\eta} - \frac{L_\theta}{2}\Big)\|\theta_{j+1}^i - \theta_j^i\|^2$$

$$\overset{(ii)}{\geq} \quad F(\lambda(\theta_j^i)) + \frac{\eta}{4}\|\mathcal{G}_\eta(\theta_j^i)\|^2 + \Big(\frac{1}{2\eta} - L_\theta\Big)\|\theta_{j+1}^i - \theta_j^i\|^2 - \Big(\frac{\eta}{2} + \frac{1}{2L_\theta}\Big)\|\nabla_\theta F(\lambda(\theta_j^i)) - g_j^i\|^2$$

where (i) is due to (12); (ii) is due to

$$\langle \nabla_\theta F(\lambda(\theta_j^i)) - g_j^i, \theta_{j+1}^i - \theta_j^i \rangle \geq -\frac{1}{2L_\theta}\|\nabla_\theta F(\lambda(\theta_j^i)) - g_j^i\|^2 - \frac{L_\theta}{2}\|\theta_{j+1}^i - \theta_j^i\|^2$$

and adding

$$0 \geq \frac{\eta}{4}\|\mathcal{G}_\eta(\theta_j^i)\|^2 - \frac{1}{2\eta}\cdot\|\theta_{j+1}^i - \theta_j^i\|^2 - \frac{\eta}{2}\cdot\|g_j^i - \nabla_\theta F(\lambda(\theta_j^i))\|^2$$

to both sides of (i). Taking expectation on both sides of the above inequality proves the lemma. $\square$

## D  Proof of Lemma 5.6

*Proof.* Due to the parameterization form (1), for any $\theta_1, \theta_2$ and any state-action pair $(s, a)$, we have

$$\frac{\pi_{\theta_2}(a|s)}{\pi_{\theta_1}(a|s)} = \frac{\exp\{\psi(s, a; \theta_2)\}}{\exp\{\psi(s, a; \theta_1)\}} \cdot \frac{\sum_{a'} \exp\{\psi(s, a'; \theta_1)\}}{\sum_{a'} \exp\{\psi(s, a'; \theta_2)\}}$$

$$\leq \exp\{\psi(s, a; \theta_2) - \psi(s, a; \theta_1)\} \cdot \max_{a'} \exp\{\psi(s, a'; \theta_1) - \psi(s, a'; \theta_2)\}$$

$$\leq \exp\{2\ell_\psi \cdot \|\theta_1 - \theta_2\|\}$$

As a result, by definition, for any $t \in \{0, 1, ..., H-1\}$, the importance sampling weight is

$$\omega_t(\tau|\theta_1, \theta_2) = \prod_{t'=0}^{t} \frac{\pi_{\theta_2}(a_{t'}|s_{t'})}{\pi_{\theta_1}(a_{t'}|s_{t'})} \leq \exp\{2(t+1)\ell_\psi \cdot \|\theta_1 - \theta_2\|\}.$$

$$\square$$

## E  A few supporting lemmas

First, we would like to introduce the lemma that describes the properties of the off-policy sampling estimators. For the ease of discussion, let us define the occupancy measure of a $H$-horizon truncated trajectory as

$$\lambda_H(s, a; \theta) := \sum_{t=0}^{H-1} \gamma^t \cdot \mathbb{P}\Big(s_t = s, a_t = a \,\big|\, \pi_\theta, s_0 \sim \xi\Big), \tag{20}$$

for $\forall(s, a) \in \mathcal{S} \times \mathcal{A}$. Then for any vector $r$, we have

$$[\nabla_\theta \lambda_H(\theta)]^\top r = \mathbb{E}\Bigg[\sum_{t=0}^{H-1} \gamma^t \cdot r(s_t, a_t) \cdot \Big(\sum_{t'=0}^{t} \nabla_\theta \log \pi_\theta(a_{t'}|s_{t'})\Big)\Big|\pi_\theta, s_0 \sim \xi\Bigg]. \tag{21}$$

Next, let us focus on the off-policy estimators.

**Proposition E.1.** *Let* $\tau = \{s_0, a_0, s_1, a_1, \cdots, s_{H-1}, a_{H-1}\}$ *be sampled from the behavioral policy* $\pi_{\theta_1}$. *Then for the target policy* $\pi_{\theta_2}$, *it holds that*

$$\mathbb{E}_{\tau \sim \pi_{\theta_1}} \left[ \widehat{\lambda}_\omega(\tau|\theta_1, \theta_2) \right] = \lambda_H(\theta_2) \qquad and \qquad \mathbb{E}_{\tau \sim \pi_{\theta_1}} [\widehat{g}_\omega(\tau|\theta_1, \theta_2, r)] = [\nabla_\theta \lambda_H(\theta_2)]^\top r.$$

*The above equations also hold in case* $\theta_1 = \theta_2$.

*Proof.* First, let us define $\tau_t = \{s_0, a_0, ..., s_t, a_t\}$ as the truncated trajectory of $\tau$ with length $t + 1$. Then we can write

$$p(\tau_t|\pi_\theta) = \xi(s_0)\pi_\theta(a_0|s_0) \cdot \prod_{t'=1}^{t} P(s_{t'}|s_{t'-1}, a_{t'-1})\pi_\theta(a_{t'}|s_{t'}).$$

Then we also have $\omega_t(\tau|\theta_1, \theta_2) = \frac{p(\tau_t|\pi_{\theta_2})}{p(\tau_t|\pi_{\theta_1})}$. Consequently, we have

$$
\begin{aligned}
&\mathbb{E}_{\tau \sim \pi_{\theta_1}} \left[ \widehat{\lambda}_\omega(\tau|\theta_1, \theta_2) \right] \\
=\ & \sum_\tau p(\tau|\pi_{\theta_1}) \cdot \left( \sum_{t=0}^{H-1} \gamma^t \cdot \omega_t(\tau|\theta_1, \theta_2) \cdot \mathbf{e}_{s_t a_t} \right) \\
=\ & \sum_\tau p(\tau|\pi_{\theta_1}) \cdot \left( \sum_{t=0}^{H-1} \gamma^t \cdot \frac{p(\tau_t|\pi_{\theta_2})}{p(\tau_t|\pi_{\theta_1})} \cdot \mathbf{e}_{s_t a_t} \right) \\
=\ & \sum_\tau \sum_{t=0}^{H-1} \gamma^t \cdot p(\tau_t|\pi_{\theta_2}) \left( \prod_{t'=t+1}^{H-1} P(s_{t'}|s_{t'-1}, a_{t'-1})\pi_{\theta_1}(a_{t'}|s_{t'}) \right) \cdot \mathbf{e}_{s_t a_t} \\
=\ & \sum_{\tau_t} \gamma^t \cdot p(\tau_t|\pi_{\theta_2}) \sum_{\{s_{t'}, a_{t'}\}_{t'=t+1}^{H-1}} \left( \prod_{t'=t+1}^{H-1} P(s_{t'}|s_{t'-1}, a_{t'-1})\pi_{\theta_1}(a_{t'}|s_{t'}) \right) \cdot \mathbf{e}_{s_t a_t} \\
\overset{(i)}{=}\ & \sum_{\tau_t} \gamma^t \cdot p(\tau_t|\pi_{\theta_2}) \sum_{\{s_{t'}, a_{t'}\}_{t'=t+1}^{H-1}} \left( \prod_{t'=t+1}^{H-1} P(s_{t'}|s_{t'-1}, a_{t'-1})\pi_{\theta_2}(a_{t'}|s_{t'}) \right) \cdot \mathbf{e}_{s_t a_t} \\
=\ & \sum_\tau \sum_{t=0}^{H-1} \gamma^t \cdot p(\tau_t|\pi_{\theta_2}) \left( \prod_{t'=t+1}^{H-1} P(s_{t'}|s_{t'-1}, a_{t'-1})\pi_{\theta_2}(a_{t'}|s_{t'}) \right) \cdot \mathbf{e}_{s_t a_t} \\
=\ & \sum_\tau p(\tau|\pi_{\theta_2}) \cdot \left( \sum_{t=0}^{H-1} \gamma^t \cdot \mathbf{e}_{s_t a_t} \right) \\
=\ & \lambda_H(\theta_2),
\end{aligned}
$$

where (i) is due to the fact that $\sum_{\{s_{t'}, a_{t'}\}_{t'=t+1}^{H-1}} \left( \prod_{t'=t+1}^{H-1} P(s_{t'}|s_{t'-1}, a_{t'-1})\pi_\theta(a_{t'}|s_{t'}) \right) \equiv 1$ for any policy $\pi_\theta$. Similarly, we also have

$$\mathbb{E}\left[\widehat{g}_\omega(\tau|\theta_1, \theta_2, r)\right] = [\nabla_\theta \lambda_H(\theta_2)]^\top r.$$

$\square$

Next, we introduce the Lipschitz continuity of these estimators.

**Lemma E.2.** *Let* $\tau = \{s_0, a_0, ..., s_{H-1}, a_{H-1}\}$ *be an arbitrary trajectory. Then the following inequalities hold true.*

(i). *For* $\forall \theta$, *and* $\forall r_1, r_2$, *it holds that* $\|\widehat{g}(\tau|\theta, r_1) - \widehat{g}(\tau|\theta, r_2)\| \le \frac{2\ell_\psi}{(1-\gamma)^2} \cdot \|r_1 - r_2\|_\infty$.

(ii). *For* $\forall \theta_1, \theta_2$, *and* $\forall r$, *it holds that* $\|\widehat{g}(\tau|\theta_1, r) - \widehat{g}(\tau|\theta_2, r)\| \le \frac{2(\ell_\psi^2 + L_\psi) \cdot \|r\|_\infty}{(1-\gamma)^2} \cdot \|\theta_1 - \theta_2\|$.

*Proof.* For the first inequality, we have

$$
\begin{aligned}
\|\widehat{g}(\tau|\theta, r_1) - \widehat{g}(\tau|\theta, r_2)\| &= \left\| \sum_{t=0}^{H-1} \gamma^t \cdot (r_1(s_t, a_t) - r_2(s_t, a_t)) \cdot \Big( \sum_{t'=0}^{t} \nabla_\theta \log \pi_\theta(a_{t'}|s_{t'}) \Big) \right\| \\
&\leq \sum_{t=0}^{H-1} \gamma^t (t+1) \cdot 2\ell_\psi \cdot \|r_1 - r_2\|_\infty \\
&\leq \frac{2\ell_\psi \cdot \|r_1 - r_2\|_\infty}{(1-\gamma)^2}.
\end{aligned}
$$

For the second inequality, we have

$$
\begin{aligned}
\|\widehat{g}(\tau|\theta_1, r) - \widehat{g}(\tau|\theta_2, r)\| &= \left\| \sum_{t=0}^{H-1} \gamma^t \cdot r(s_t, a_t) \cdot \Big( \sum_{t'=0}^{t} \nabla_\theta \log \pi_{\theta_1}(a_{t'}|s_{t'}) - \nabla_\theta \log \pi_{\theta_2}(a_{t'}|s_{t'}) \Big) \right\| \\
&\leq \sum_{t=0}^{H-1} \gamma^t (t+1) \cdot 2(\ell_\psi^2 + L_\psi) \cdot \|\theta_1 - \theta_2\| \cdot \|r\|_\infty \\
&\leq \frac{2(\ell_\psi^2 + L_\psi) \cdot \|r\|_\infty}{(1-\gamma)^2} \cdot \|\theta_1 - \theta_2\|.
\end{aligned}
$$

Hence we complete the proof. $\qquad \square$

Finally, we provide the following lemma to characterize the property of the truncated occupancy measure.

**Lemma E.3.** *For any $H$, the following inequality holds that*

$$
\left\| \nabla_\theta F(\lambda_H(\theta)) - \nabla_\theta F(\lambda(\theta)) \right\|^2 \leq \left( \frac{8\ell_\psi^2 \cdot L_\lambda^2}{(1-\gamma)^6} + 16\ell_\psi^2 \ell_{\lambda,\infty}^2 \left( \frac{(H+1)^2}{(1-\gamma)^2} + \frac{1}{(1-\gamma)^4} \right) \right) \cdot \gamma^{2H}.
$$

*Proof.* First, by triangle inequality, we get

$$
\begin{aligned}
& \left\| \nabla_\theta F(\lambda_H(\theta)) - \nabla_\theta F(\lambda(\theta)) \right\|^2 \\
={}& \left\| [\nabla_\theta \lambda_H(\theta)]^\top \nabla_\lambda F(\lambda_H(\theta)) - [\nabla_\theta \lambda(\theta)]^\top \nabla_\lambda F(\lambda(\theta)) \right\|^2 \\
\leq{}& 2\left\| [\nabla_\theta \lambda_H(\theta)]^\top (\nabla_\lambda F(\lambda_H(\theta)) - \nabla_\lambda F(\lambda(\theta))) \right\|^2 \\
& +2\left\| ([\nabla_\theta \lambda_H(\theta)]^\top - [\nabla_\theta \lambda(\theta)]^\top) \nabla_\lambda F(\lambda(\theta)) \right\|^2.
\end{aligned}
$$

For the first term, we have

$$
\begin{aligned}
& \left\| [\nabla_\theta \lambda_H(\theta_j^i)]^\top (\nabla_\lambda F(\lambda_H(\theta_j^i)) - \nabla_\lambda F(\lambda(\theta_j^i))) \right\|^2 \\
\overset{(a)}{\leq}{}& \frac{4\ell_\psi^2}{(1-\gamma)^4} \|\nabla_\lambda F(\lambda_H(\theta_j^i)) - \nabla_\lambda F(\lambda(\theta_j^i))\|_\infty^2 \\
\overset{(b)}{\leq}{}& \frac{4\ell_\psi^2 \cdot L_{\lambda,\infty}^2}{(1-\gamma)^4} \|\lambda_H(\theta_j^i) - \lambda(\theta_j^i)\|_1^2 \\
\leq{}& \frac{4\ell_\psi^2 \cdot L_{\lambda,\infty}^2}{(1-\gamma)^6} \cdot \gamma^{2H}.
\end{aligned}
$$

In the above argument, (a) follows the argument of (18) and (b) is due to Assumption 5.2. For the second term, we have

$$
\left\| [\nabla_\theta \lambda_H(\theta) - \nabla_\theta \lambda(\theta)]^\top \nabla_\lambda F(\lambda(\theta_j^i)) \right\|^2
$$

$$
= \left\| \mathbb{E}\left[ \sum_{t=H}^{+\infty} \gamma^t \cdot \nabla_{\lambda_{s_t a_t}} F(\lambda(\theta_j^i)) \cdot \left( \sum_{t'=0}^t \nabla_\theta \log \pi_\theta(a_{t'}|s_{t'}) \right) \right] \right\|^2
$$

$$
\leq \left( \sum_{t=H}^{+\infty} \gamma^t \cdot 2(t+1)\ell_\psi \cdot \ell_{\lambda,\infty} \right)^2
$$

$$
\leq 8\ell_\psi^2 \ell_{\lambda,\infty}^2 \left( \frac{(H+1)^2}{(1-\gamma)^2} + \frac{1}{(1-\gamma)^4} \right) \cdot \gamma^{2H}.
$$

Combining the above inequalities proves the lemma. $\qquad\square$

# F Proof of Lemma 5.8

## F.1 Bounding the variance of $\lambda_j^i$

To prove this lemma, the first step is to bound the variance of the occupancy estimator $\lambda_j^i$, which is shown in the following supporting lemma.

**Lemma F.1.** *For the occupancy estimators $\lambda_j^i$, we have*

$$
\mathbb{E}\left[ \left\| \lambda_j^i - \lambda_H(\theta_j^i) \right\|^2 \right] \leq \frac{1}{N(1-\gamma)^2} + \frac{2H(8\ell_\psi^2 + L_\psi)(e^{4H\ell_\psi \delta} + 1)}{(1-\gamma)^3 \cdot B} \cdot \sum_{j'=1}^j \mathbb{E}\left[ \left\| \theta_{j'-1}^i - \theta_{j'}^i \right\|^2 \right],
$$

*where $\lambda_H$ is defined in Appendix E.*

*Proof.* First, let us prove this lemma for the case $j = 0$. Note that $\lambda_0^i = \frac{1}{N} \sum_{\tau \in \mathcal{N}_i} \widehat{\lambda}(\tau|\theta_0^i)$, by the independence of the trajectories and every $\tau \in \mathcal{N}_i$ is sampled under $\pi_{\theta_0^i}$, we have

$$
\mathbb{E}\left[ \left\| \lambda_0^i - \lambda_H(\theta_0^i) \right\|^2 \right] = \mathbb{E}\left[ \left\| \frac{1}{N} \sum_{\tau \in \mathcal{N}_i} \widehat{\lambda}(\tau|\theta_0^i) - \lambda_H(\theta_0^i) \right\|^2 \right]
$$

$$
\overset{(a)}{=} \frac{1}{N} \cdot \mathbb{E}\left[ \left\| \widehat{\lambda}(\tau|\theta_0^i) - \lambda_H(\theta_0^i) \right\|^2 \right]
$$

$$
\overset{(b)}{\leq} \frac{1}{N} \cdot \mathbb{E}\left[ \left\| \widehat{\lambda}(\tau|\theta_0^i) \right\|^2 \right]
$$

$$
\overset{(c)}{\leq} (1-\gamma)^{-2} \cdot N^{-1},
$$

where (a) is due to $\lambda_H(\theta_0^i) = \mathbb{E}\left[ \widehat{\lambda}(\tau|\theta_0^i) \right]$, see Proposition E.1, and the fact that the trajectories in $\mathcal{N}_i$ are independently sampled; (b) is because $\mathrm{Var}(X) \leq \mathbb{E}\left[ \|X\|^2 \right]$ for any random vector $X$; (c) is because $\left\| \widehat{\lambda}(\tau|\theta_0^i) \right\| \leq (1-\gamma)^{-1}$ w.p. 1. Next, let us prove the lemma for the case $j \geq 1$, where each trajectory $\tau \in \mathcal{B}_j^i$ is sampled under policy $\pi_{\theta_j^i}$.

$$
\mathbb{E}\left[ \|\lambda_j^i - \lambda_H(\theta_j^i)\|^2 \right] \tag{22}
$$

$$
= \mathbb{E}\left[ \left\| \frac{1}{B} \sum_{\tau \in \mathcal{B}_j^i} \left( \widehat{\lambda}(\tau|\theta_j^i) - \widehat{\lambda}_\omega\left( \tau|\theta_j^i, \theta_{j-1}^i \right) \right) + \lambda_{j-1}^i - \lambda_H(\theta_{j-1}^i) + \lambda_H(\theta_{j-1}^i) - \lambda_H(\theta_j^i) \right\|^2 \right]
$$

$$
= \mathbb{E}\left[ \left\| \frac{1}{B} \sum_{\tau \in \mathcal{B}_j^i} \left( \widehat{\lambda}(\tau|\theta_j^i) - \widehat{\lambda}_\omega\left( \tau|\theta_j^i, \theta_{j-1}^i \right) \right) + \lambda_H(\theta_{j-1}^i) - \lambda_H(\theta_j^i) \right\|^2 \right] + \mathbb{E}\left[ \left\| \lambda_{j-1}^i - \lambda_H(\theta_{j-1}^i) \right\|^2 \right]
$$

$$
+ 2\mathbb{E}\left[ \left\langle \frac{1}{B} \sum_{\tau \in \mathcal{B}_j^i} \left( \widehat{\lambda}(\tau|\theta_j^i) - \widehat{\lambda}_\omega\left( \tau|\theta_j^i, \theta_{j-1}^i \right) \right) + \lambda_H(\theta_{j-1}^i) - \lambda_H(\theta_j^i), \lambda_{j-1}^i - \lambda_H(\theta_{j-1}^i) \right\rangle \right]
$$

Note that

$$\mathbb{E}_{\tau \sim \pi_{\theta_j^i}}\left[\frac{1}{B}\sum_{\tau \in \mathcal{B}_j^i}\widehat{\lambda}(\tau|\theta_j^i) - \lambda_H(\theta_j^i)\right] = \mathbb{E}_{\tau \sim \pi_{\theta_j^i}}\left[\frac{1}{B}\sum_{\tau \in \mathcal{B}_j^i}\widehat{\lambda}_\omega\left(\tau|\theta_j^i, \theta_{j-1}^i\right) - \lambda_H(\theta_{j-1}^i)\right] = 0.$$

For the first term of (22), we have

$$\mathbb{E}\left[\left\|\frac{1}{B}\sum_{\tau \in \mathcal{B}_j^i}\left(\widehat{\lambda}(\tau|\theta_j^i) - \widehat{\lambda}_\omega\left(\tau|\theta_j^i, \theta_{j-1}^i\right)\right) + \lambda_H(\theta_{j-1}^i) - \lambda_H(\theta_j^i)\right\|^2\right] \tag{23}$$

$$\overset{(a)}{=} \frac{1}{B}\cdot\mathbb{E}\left[\left\|\widehat{\lambda}(\tau|\theta_j^i) - \widehat{\lambda}_\omega\left(\tau|\theta_j^i, \theta_{j-1}^i\right) + \lambda_H(\theta_{j-1}^i) - \lambda_H(\theta_j^i)\right\|^2\right]$$

$$\overset{(b)}{\le} \frac{1}{B}\cdot\mathbb{E}\left[\left\|\widehat{\lambda}(\tau|\theta_j^i) - \widehat{\lambda}_\omega\left(\tau|\theta_j^i, \theta_{j-1}^i\right)\right\|^2\right]$$

$$\overset{(c)}{=} \frac{1}{B}\cdot\mathbb{E}\left[\left\|\sum_{t=0}^{H-1}\gamma^t\cdot\left(1 - \omega_t(\tau|\theta_j^i, \theta_{j-1}^i)\right)\cdot\mathbf{e}_{s_t a_t}\right\|^2\right]$$

$$\overset{(d)}{\le} \frac{H}{B}\cdot\sum_{t=0}^{H-1}\gamma^t\cdot\mathrm{Var}\left(\omega_t(\tau|\theta_j^i, \theta_{j-1}^i)\right)$$

$$\overset{(e)}{\le} \frac{H\cdot\mathbb{E}\left[\|\theta_{j-1}^i - \theta_j^i\|^2\right]}{B}\cdot\sum_{t=0}^{H-1}\gamma^t\cdot(t+1)\left(8(t+1)\ell_\psi^2 + 2(\ell_\psi^2 + L_\psi)\right)(e^{4H\ell_\psi\delta} + 1)$$

$$\le \frac{H\cdot\mathbb{E}\left[\|\theta_{j-1}^i - \theta_j^i\|^2\right]}{B}\cdot\sum_{t=0}^{H-1}\gamma^t\cdot 2(t+1)(t+2)\left(8\ell_\psi^2 + L_\psi\right)(e^{4H\ell_\psi\delta} + 1)$$

$$\le \frac{2H(8\ell_\psi^2 + L_\psi)(e^{4H\ell_\psi\delta} + 1)}{(1-\gamma)^3\cdot B}\cdot\mathbb{E}\left[\|\theta_{j-1}^i - \theta_j^i\|^2\right]$$

where (a) is due to the unbiasedness of the difference estimator; (b) is because $\mathrm{Var}(X) \le \mathbb{E}[\|X\|^2]$ for any random vector $X$; (c) is due to the definition (6); (d) utilizes the inequality that $(\sum_{h=1}^H x_h)^2 \le H\sum_{h=1}^H x_h^2$ and (e) is due to Lemma 5.7. Now, substituting (23) into (22) yields

$$\mathbb{E}\left[\|\lambda_j^i - \lambda_H(\theta_j^i)\|^2\right] \le \mathbb{E}\left[\|\lambda_{j-1}^i - \lambda_H(\theta_{j-1}^i)\|^2\right] + \frac{2H(8\ell_\psi^2 + L_\psi)(e^{4H\ell_\psi\delta} + 1)}{(1-\gamma)^3\cdot B}\cdot\mathbb{E}\left[\|\theta_{j-1}^i - \theta_j^i\|^2\right].$$

Recursively summing the above inequalities up yields

$$\mathbb{E}\left[\|\lambda_j^i - \lambda_H(\theta_j^i)\|^2\right] \le \mathbb{E}\left[\|\lambda_0^i - \lambda_H(\theta_0^i)\|^2\right] + \frac{2H(8\ell_\psi^2 + L_\psi)(e^{4H\ell_\psi\delta} + 1)}{(1-\gamma)^3\cdot B}\cdot\sum_{j'=1}^{j}\mathbb{E}\left[\|\theta_{j-1}^i - \theta_j^i\|^2\right]$$

$$\le (1-\gamma)^{-2}N^{-1} + \frac{2H(8\ell_\psi^2 + L_\psi)(e^{4H\ell_\psi\delta} + 1)}{(1-\gamma)^3\cdot B}\cdot\sum_{j'=1}^{j}\mathbb{E}\left[\|\theta_{j-1}^i - \theta_j^i\|^2\right].$$

Combining the case for $j = 0$ and $j \ge 1$ proves the lemma. $\qquad\square$

### F.2 Proof of Lemma 5.8

First, we present a more detailed version of Lemma 5.8, which contains the expressions of the constants $C_i, i = 1, ..., 4$.

**Lemma F.2.** *For the PG estimators $g_j^i$, we have*

$$\mathbb{E}\left[\left\|g_j^i - \nabla_\theta F(\lambda(\theta_j^i))\right\|^2\right] \le \frac{C_1}{N} + C_2\gamma^{2H} + \frac{C_3}{B}\cdot\sum_{j'=1}^{j}\mathbb{E}\left[\|\theta_{j'-1}^i - \theta_{j'}^i\|^2\right] + C_4\mathbb{E}\left[\|\theta_{j-1}^i - \theta_j^i\|^2\right]$$

*where the constants $C_1, C_2, C_3$ and $C_4$ are defined as*

$$C_1 = \frac{112\ell_\psi^2\cdot L_\lambda^2}{(1-\gamma)^6} + \frac{12\ell_{\lambda,\infty}^2}{(1-\gamma)^4}, \qquad C_2 = \frac{32\ell_\psi^2\cdot L_\lambda^2}{(1-\gamma)^6} + 64\ell_\psi^2\ell_{\lambda,\infty}^2\left(\frac{(H+1)^2}{(1-\gamma)^2} + \frac{1}{(1-\gamma)^4}\right)$$

$$C_3 = \frac{48(\ell_\psi + L_\psi)^2 \ell_{\lambda,\infty}^2}{(1-\gamma)^4} + \frac{96H\ell_{\lambda,\infty}^2\left(8\ell_\psi^2 + L_\psi\right)(e^{4H\ell_\psi\delta}+1)}{(1-\gamma)^5} \cdot \left(12\ell_{\lambda,\infty}^2 + \frac{4L_\lambda^2}{3(1-\gamma)^2}\right)$$

$$C_4 = \frac{32H\ell_\psi^2 L_\lambda^2 (8\ell_\psi^2 + L_\psi)(e^{4H\ell_\psi\delta}+1)}{(1-\gamma)^7}.$$

*Proof.* The proof of this lemma is very lengthy, we will separate it into several steps:

**Step 1.** Show for any $j \geq 1$ that

$$\mathbb{E}\left[\left\|g_j^i - \left[\nabla_\theta \lambda_H(\theta_j^i)\right]^\top r_{j-1}^i\right\|^2\right] - \mathbb{E}\left[\left\|g_0^i - \left[\nabla_\theta \lambda_H(\theta_0^i)\right]^\top r_0^i\right\|^2\right] \tag{24}$$

$$\leq \left(\frac{12(\ell_\psi + L_\psi)^2 \ell_{\lambda,\infty}^2}{(1-\gamma)^4} + \frac{24H\ell_{\lambda,\infty}^2\left(8\ell_\psi^2 + L_\psi\right)(e^{4H\ell_\psi\delta}+1)}{(1-\gamma)^5}\cdot\left(12\ell_{\lambda,\infty}^2 + \frac{L_\lambda^2}{(1-\gamma)^2}\right)\right)\cdot\frac{\sum_{j'=1}^j \mathbb{E}\left[\|\theta_{j'}^i - \theta_{j'-1}^i\|^2\right]}{B}.$$

**Step 2.** Show for $j = 0$ that

$$\mathbb{E}\left[\left\|g_0^i - \left[\nabla_\theta \lambda_H(\theta_0^i)\right]^\top r_0^i\right\|^2\right] \leq \frac{24\ell_\psi^2 \cdot L_\lambda^2}{(1-\gamma)^6 \cdot N} + \frac{3\ell_{\lambda,\infty}^2}{(1-\gamma)^4 \cdot N}. \tag{25}$$

**Step 3.** Find the bound for the final mean squared error $\mathbb{E}[\|g_j^i - \nabla_\theta F(\lambda(\theta_j^i))\|^2]$.

Compared to Step 1 and 2, Step 3 is relatively simple so we place the proof of Step 1 and Step 2 to Appendix F.3 and F.4 respectively. Given these inequalities, we have

$$\mathbb{E}\left[\left\|g_j^i - \nabla_\theta F(\lambda(\theta_j^i))\right\|^2\right]$$
$$= \mathbb{E}\Big[\big\|g_j^i - \left[\nabla_\theta \lambda_H(\theta_j^i)\right]^\top r_{j-1}^i + \left[\nabla_\theta \lambda_H(\theta_j^i)\right]^\top r_{j-1}^i - \left[\nabla_\theta \lambda_H(\theta_j^i)\right]^\top r_j^i$$
$$+ \left[\nabla_\theta \lambda_H(\theta_j^i)\right]^\top r_j^i - \nabla_\theta F(\lambda_H(\theta_j^i)) + \nabla_\theta F(\lambda_H(\theta_j^i)) - \nabla_\theta F(\lambda(\theta_j^i))\big\|^2\Big]$$
$$\leq 4\mathbb{E}\left[\left\|g_j^i - \left[\nabla_\theta \lambda_H(\theta_j^i)\right]^\top r_{j-1}^i\right\|^2\right] + 4\mathbb{E}\left[\left\|\left[\nabla_\theta \lambda_H(\theta_j^i)\right]^\top r_{j-1}^i - \left[\nabla_\theta \lambda_H(\theta_j^i)\right]^\top r_j^i\right\|^2\right]$$
$$+ 4\mathbb{E}\left[\left\|\left[\nabla_\theta \lambda_H(\theta_j^i)\right]^\top r_j^i - \nabla_\theta F(\lambda_H(\theta_j^i))\right\|^2\right] + 4\mathbb{E}\left[\left\|\nabla_\theta F(\lambda_H(\theta_j^i)) - \nabla_\theta F(\lambda(\theta_j^i))\right\|^2\right]$$

The first term is given by (24) and (25). The second term is similar to the proof of (18) and (29), which gives

$$\mathbb{E}\left[\left\|\left[\nabla_\theta \lambda_H(\theta_j^i)\right]^\top r_{j-1}^i - \left[\nabla_\theta \lambda_H(\theta_j^i)\right]^\top r_j^i\right\|^2\right]$$
$$\leq \frac{4\ell_\psi^2}{(1-\gamma)^4} \cdot \mathbb{E}\left[\|r_{j-1}^i - r_j^i\|_\infty^2\right]$$
$$= \frac{4\ell_\psi^2}{(1-\gamma)^4} \cdot \mathbb{E}\left[\|\nabla_\lambda F(\lambda_{j-1}^i) - \nabla_\lambda F(\lambda_j^i)\|_\infty^2\right]$$
$$\leq \frac{4\ell_\psi^2 \cdot L_\lambda^2}{(1-\gamma)^4} \cdot \mathbb{E}\left[\|\lambda_{j-1}^i - \lambda_j^i\|^2\right]$$
$$\leq \frac{4\ell_\psi^2 \cdot L_\lambda^2}{(1-\gamma)^4} \cdot \frac{2H(8\ell_\psi^2 + L_\psi)(e^{4H\ell_\psi\delta}+1)}{(1-\gamma)^3} \cdot \mathbb{E}\left[\|\theta_j^i - \theta_{j-1}^i\|^2\right].$$

For the third term,

$$\mathbb{E}\left[\left\|\left[\nabla_\theta \lambda_H(\theta_j^i)\right]^\top r_j^i - \nabla_\theta F(\lambda_H(\theta_j^i))\right\|^2\right]$$
$$= \mathbb{E}\left[\left\|\left[\nabla_\theta \lambda_H(\theta_j^i)\right]^\top \nabla_\lambda F(\lambda_j^i) - \left[\nabla_\theta \lambda_H(\theta_j^i)\right]^\top \nabla_\lambda F(\lambda_H(\theta_j^i))\right\|^2\right]$$
$$\leq \frac{4\ell_\psi^2 \cdot L_\lambda^2}{(1-\gamma)^4} \cdot \mathbb{E}\left[\|\lambda_j^i - \lambda_H(\theta_j^i)\|^2\right]$$

where $\mathbb{E}\left[\|\lambda_j^i - \lambda_H(\theta_j^i)\|^2\right]$ is given by Lemma F.1. For the last term, Lemma E.3 indicates that

$$\left\|\nabla_\theta F(\lambda_H(\theta_j^i)) - \nabla_\theta F(\lambda(\theta_j^i))\right\|^2 \leq \left(\frac{8\ell_\psi^2 \cdot L_\lambda^2}{(1-\gamma)^6} + 16\ell_\psi^2 \ell_{\lambda,\infty}^2 \left(\frac{(H+1)^2}{(1-\gamma)^2} + \frac{1}{(1-\gamma)^4}\right)\right) \cdot \gamma^{2H}.$$

Combining all the above inequalities, we have the final result:

$$\mathbb{E}\left[\left\|g_j^i - \nabla_\theta F(\lambda(\theta_j^i))\right\|^2\right] \leq \frac{C_1}{N} + C_2\gamma^{2H} + \frac{C_3}{B} \cdot \sum_{j'=1}^{j} \mathbb{E}\left[\|\theta_{j'-1}^i - \theta_{j'}^i\|^2\right] + C_4\mathbb{E}\left[\|\theta_{j-1}^i - \theta_j^i\|^2\right].$$

$\square$

## F.3    Proof of Step 1

*Proof.* Consider the $j$-th step of the $i$-th epoch where the trajectories $\tau \in \mathcal{B}_j^i$ are sampled under policy $\pi_{\theta_j^i}$. Similar to the analysis of Lemma F.1, we have

$$\mathbb{E}\left[\left\|g_j^i - \left[\nabla_\theta \lambda_H(\theta_j^i)\right]^\top r_{j-1}^i\right\|^2\right] - \mathbb{E}\left[\left\|g_{j-1}^i - \left[\nabla_\theta \lambda_H(\theta_{j-1}^i)\right]^\top r_{j-2}^i\right\|^2\right] \tag{26}$$

$$= \mathbb{E}\bigg[\bigg\|\frac{1}{B}\sum_{\tau \in \mathcal{B}_j^i}\left(\widehat{g}(\tau|\theta_j^i, r_{j-1}^i) - \widehat{g}_\omega(\tau|\theta_j^i, \theta_{j-1}^i, r_{j-2}^i)\right) + g_{j-1}^i - \left[\nabla_\theta \lambda_H(\theta_{j-1}^i)\right]^\top r_{j-2}^i$$

$$+ \left[\nabla_\theta \lambda_H(\theta_{j-1}^i)\right]^\top r_{j-2}^i - \left[\nabla_\theta \lambda_H(\theta_j^i)\right]^\top r_{j-1}^i\bigg\|^2\bigg] - \mathbb{E}\left[\left\|g_{j-1}^i - \left[\nabla_\theta \lambda_H(\theta_{j-1}^i)\right]^\top r_{j-2}^i\right\|^2\right]$$

$$= \mathbb{E}\bigg[\bigg\|\frac{1}{B}\sum_{\tau \in \mathcal{B}_j^i}\left(\widehat{g}(\tau|\theta_j^i, r_{j-1}^i) - \widehat{g}_\omega(\tau|\theta_j^i, \theta_{j-1}^i, r_{j-2}^i)\right) + \left[\nabla_\theta \lambda_H(\theta_{j-1}^i)\right]^\top r_{j-2}^i - \left[\nabla_\theta \lambda_H(\theta_j^i)\right]^\top r_{j-1}^i\bigg\|^2\bigg]$$

$$+ 2\mathbb{E}\bigg[\bigg\langle\frac{1}{B}\sum_{\tau \in \mathcal{B}_j^i}\left(\widehat{g}(\tau|\theta_j^i, r_{j-1}^i) - \widehat{g}_\omega(\tau|\theta_j^i, \theta_{j-1}^i, r_{j-2}^i)\right) + \left[\nabla_\theta \lambda_H(\theta_{j-1}^i)\right]^\top r_{j-2}^i$$

$$- \left[\nabla_\theta \lambda_H(\theta_j^i)\right]^\top r_{j-1}^i \,,\; g_{j-1}^i - \left[\nabla_\theta \lambda_H(\theta_{j-1}^i)\right]^\top r_{j-2}^i\bigg\rangle\bigg].$$

Let $\mathcal{F}_{j-1}^i$ be the sigma algebra generated by the randomness until (including) the trajectory batch $\mathcal{B}_j^i$. Then we have

$$\mathbb{E}\left[\left\langle\frac{1}{B}\sum_{\tau \in \mathcal{B}_j^i}\widehat{g}(\tau|\theta_j^i, r_{j-1}^i) - \left[\nabla_\theta \lambda_H(\theta_j^i)\right]^\top r_{j-1}^i \,,\; g_{j-1}^i - \left[\nabla_\theta \lambda_H(\theta_{j-1}^i)\right]^\top r_{j-2}^i\right\rangle \bigg| \mathcal{F}_{j-1}^i\right]$$

$$= \left\langle\frac{1}{B}\sum_{\tau \in \mathcal{B}_j^i}\mathbb{E}\left[\widehat{g}(\tau|\theta_j^i, r_{j-1}^i) \big| \mathcal{F}_{j-1}^i\right] - \left[\nabla_\theta \lambda_H(\theta_j^i)\right]^\top r_{j-1}^i \,,\; g_{j-1}^i - \left[\nabla_\theta \lambda_H(\theta_{j-1}^i)\right]^\top r_{j-2}^i\right\rangle$$

$$= 0.$$

Similarly, we have

$$\mathbb{E}\left[\left\langle -\frac{1}{B}\sum_{\tau \in \mathcal{B}_j^i}\widehat{g}_\omega(\tau|\theta_j^i, \theta_{j-1}^i, r_{j-2}^i) + \left[\nabla_\theta \lambda_H(\theta_{j-1}^i)\right]^\top r_{j-2}^i \,,\; g_{j-1}^i - \left[\nabla_\theta \lambda_H(\theta_{j-1}^i)\right]^\top r_{j-2}^i\right\rangle \bigg| \mathcal{F}_{j-1}^i\right]$$

$$= \left\langle -\frac{1}{B}\sum_{\tau \in \mathcal{B}_j^i}\mathbb{E}\left[\widehat{g}_\omega(\tau|\theta_j^i, \theta_{j-1}^i, r_{j-2}^i) \big| \mathcal{F}_{j-1}^i\right] + \left[\nabla_\theta \lambda_H(\theta_{j-1}^i)\right]^\top r_{j-2}^i \,,\; g_{j-1}^i - \left[\nabla_\theta \lambda_H(\theta_{j-1}^i)\right]^\top r_{j-2}^i\right\rangle$$

$$= 0.$$

Substituting the above two inequalities into (26) yields

$$
\mathbb{E}\left[\left\|g_j^i - \left[\nabla_\theta \lambda_H(\theta_j^i)\right]^\top r_{j-1}^i\right\|^2\right] - \mathbb{E}\left[\left\|g_{j-1}^i - \left[\nabla_\theta \lambda_H(\theta_{j-1}^i)\right]^\top r_{j-2}^i\right\|^2\right] \tag{27}
$$

$$
= \mathbb{E}\left[\left\|\frac{1}{B}\sum_{\tau\in\mathcal{B}_j^i}\left(\widehat{g}(\tau|\theta_j^i, r_{j-1}^i) - \widehat{g}_\omega(\tau|\theta_j^i, \theta_{j-1}^i, r_{j-2}^i)\right) + \left[\nabla_\theta\lambda_H(\theta_{j-1}^i)\right]^\top r_{j-2}^i - \left[\nabla_\theta\lambda_H(\theta_j^i)\right]^\top r_{j-1}^i\right\|^2\right]
$$

$$
\leq \frac{1}{B}\cdot\mathbb{E}\left[\left\|\widehat{g}(\tau|\theta_j^i, r_{j-1}^i) - \widehat{g}_\omega(\tau|\theta_j^i, \theta_{j-1}^i, r_{j-2}^i)\right\|^2\right]
$$

$$
= \frac{1}{B}\cdot\mathbb{E}\left[\left\|\widehat{g}(\tau|\theta_j^i, r_{j-1}^i) - \widehat{g}(\tau|\theta_{j-1}^i, r_{j-1}^i) + \widehat{g}(\tau|\theta_{j-1}^i, r_{j-1}^i) - \widehat{g}(\tau|\theta_{j-1}^i, r_{j-2}^i)\right.\right.
$$

$$
\left.\left. + \widehat{g}(\tau|\theta_{j-1}^i, r_{j-2}^i) - \widehat{g}_\omega(\tau|\theta_j^i, \theta_{j-1}^i, r_{j-2}^i)\right\|^2\right]
$$

$$
\leq \frac{3}{B}\cdot\mathbb{E}\left[\left\|\widehat{g}(\tau|\theta_j^i, r_{j-1}^i) - \widehat{g}(\tau|\theta_{j-1}^i, r_{j-1}^i)\right\|^2\right] + \frac{3}{B}\cdot\mathbb{E}\left[\left\|\widehat{g}(\tau|\theta_{j-1}^i, r_{j-1}^i) - \widehat{g}(\tau|\theta_{j-1}^i, r_{j-2}^i)\right\|^2\right]
$$

$$
+ \frac{3}{B}\cdot\mathbb{E}\left[\left\|\widehat{g}(\tau|\theta_{j-1}^i, r_{j-2}^i) - \widehat{g}_\omega(\tau|\theta_j^i, \theta_{j-1}^i, r_{j-2}^i)\right\|^2\right]
$$

For the first term, we have

$$
\mathbb{E}\left[\left\|\widehat{g}(\tau|\theta_j^i, r_{j-1}^i) - \widehat{g}(\tau|\theta_{j-1}^i, r_{j-1}^i)\right\|^2\right] \tag{28}
$$

$$
\overset{(a)}{\leq} \frac{4(\ell_\psi^2 + L_\psi)^2}{(1-\gamma)^4}\cdot\mathbb{E}\left[\|r_{j-1}^i\|_\infty^2\cdot\|\theta_j^i - \theta_{j-1}^i\|^2\right]
$$

$$
\overset{(b)}{\leq} \frac{4(\ell_\psi^2 + L_\psi)^2\cdot\ell_{\lambda,\infty}^2}{(1-\gamma)^4}\cdot\mathbb{E}\left[\|\theta_j^i - \theta_{j-1}^i\|^2\right],
$$

where (a) is due to Lemma E.2; and (b) is due to Assumption 5.2 indicates that $\|r_{j-1}^i\|_\infty^2 = \|\nabla_\lambda F(\lambda_{j-1}^i)\|_\infty^2 \leq \ell_{\lambda,\infty}^2$ w.p. 1. Similarly, combining Lemma E.2, Assumption 5.2 and the $\lambda_j^i$ update formula (9) yields

$$
\mathbb{E}\left[\left\|\widehat{g}(\tau|\theta_{j-1}^i, r_{j-1}^i) - \widehat{g}(\tau|\theta_{j-1}^i, r_{j-2}^i)\right\|^2\right] \tag{29}
$$

$$
\leq \frac{4\ell_\psi^2}{(1-\gamma)^4}\cdot\mathbb{E}\left[\|r_{j-1}^i - r_{j-2}^i\|_\infty^2\right]
$$

$$
= \frac{4\ell_\psi^2}{(1-\gamma)^4}\cdot\mathbb{E}\left[\|\nabla_\lambda F(\lambda_{j-1}^i) - \nabla_\lambda F(\lambda_{j-2}^i)\|_\infty^2\right]
$$

$$
\leq \frac{4\ell_\psi^2\cdot L_\lambda^2}{(1-\gamma)^4}\cdot\mathbb{E}\left[\|\lambda_{j-1}^i - \lambda_{j-2}^i\|^2\right]
$$

$$
= \frac{4\ell_\psi^2\cdot L_\lambda^2}{(1-\gamma)^4}\cdot\mathbb{E}\left[\left\|\frac{1}{B}\sum_{\tau\in\mathcal{B}_{j-1}^i}\left(\widehat{\lambda}(\tau|\theta_{j-1}^i) - \widehat{\lambda}_\omega\left(\tau|\theta_{j-1}^i, \theta_{j-2}^i\right)\right)\right\|^2\right]
$$

$$
\leq \frac{4\ell_\psi^2\cdot L_\lambda^2}{(1-\gamma)^4}\cdot\mathbb{E}\left[\left\|\widehat{\lambda}(\tau|\theta_{j-1}^i) - \widehat{\lambda}_\omega\left(\tau|\theta_{j-1}^i, \theta_{j-2}^i\right)\right\|^2\right]
$$

$$
\leq \frac{4\ell_\psi^2\cdot L_\lambda^2}{(1-\gamma)^4}\cdot\frac{2H(8\ell_\psi^2 + L_\psi)(e^{4H\ell_\psi\delta} + 1)}{(1-\gamma)^3}\cdot\mathbb{E}\left[\|\theta_{j-1}^i - \theta_{j-2}^i\|^2\right]
$$

where the last inequality is due to the analysis of (23). When $j = 1$, $r_{j-2}^i = r_{j-1}^i$ by default, and the above term is zero. To be compatible, we default $\theta_{-1}^i := \theta_0^i$. For the last term of (27), where the

trajectory $\tau$ is sampled under the behavioral policy $\pi_{\theta_j^i}$, we have

$$\mathbb{E}\left[\left\|\widehat{g}(\tau|\theta_{j-1}^i, r_{j-2}^i) - \widehat{g}_\omega(\tau|\theta_j^i, \theta_{j-1}^i, r_{j-2}^i)\right\|^2\right] \tag{30}$$

$$= \mathbb{E}\left[\left\|\sum_{t=0}^{H-1} \gamma^t \cdot \left(1 - \omega_t(\tau|\theta_j^i, \theta_{j-1}^i)\right) \cdot r_{j-2}^i(s_t, a_t) \cdot \left(\sum_{t'=0}^{t} \nabla_\theta \log \pi_{\theta_{j-1}^i}(a_{t'}|s_{t'})\right)\right\|^2\right]$$

$$\leq 4H\ell_\psi^2 \ell_{\lambda,\infty}^2 \cdot \sum_{t=0}^{H-1} \gamma^t \cdot (t+1)^2 \cdot \mathrm{Var}\left(\omega_t(\tau|\theta_j^i, \theta_{j-1}^i)\right)$$

$$\leq 4H\ell_\psi^2 \ell_{\lambda,\infty}^2 \cdot \mathbb{E}\left[\|\theta_{j-1}^i - \theta_j^i\|^2\right] \cdot \sum_{t=0}^{H-1} \gamma^t \cdot (t+1)^3 \cdot (t+2)\left(8\ell_\psi^2 + L_\psi\right)\left(e^{4H\ell_\psi\delta} + 1\right)$$

$$\leq \frac{96H\ell_\psi^2 \ell_{\lambda,\infty}^2 \left(8\ell_\psi^2 + L_\psi\right)\left(e^{4H\ell_\psi\delta} + 1\right)}{(1-\gamma)^5} \cdot \mathbb{E}\left[\|\theta_{j-1}^i - \theta_j^i\|^2\right].$$

Substituting (28), (29) and (30) into (27) yields

$$\mathbb{E}\left[\left\|g_j^i - \left[\nabla_\theta \lambda_H(\theta_j^i)\right]^\top r_{j-1}^i\right\|^2\right] - \mathbb{E}\left[\left\|g_{j-1}^i - \left[\nabla_\theta \lambda_H(\theta_{j-1}^i)\right]^\top r_{j-2}^i\right\|^2\right]$$

$$\leq \left(\frac{12(\ell_\psi + L_\psi)^2 \cdot \ell_{\lambda,\infty}^2}{(1-\gamma)^4} + \frac{288H\ell_\psi^2 \ell_{\lambda,\infty}^2\left(8\ell_\psi^2 + L_\psi\right)\left(e^{4H\ell_\psi\delta} + 1\right)}{(1-\gamma)^5}\right) \cdot \frac{\mathbb{E}\left[\|\theta_j^i - \theta_{j-1}^i\|^2\right]}{B}$$

$$+ \frac{24H\ell_\psi^2 L_\lambda^2\left(8\ell_\psi^2 + L_\psi\right)\left(e^{4H\ell_\psi\delta} + 1\right)}{(1-\gamma)^7} \cdot \frac{\mathbb{E}\left[\|\theta_{j-1}^i - \theta_{j-2}^i\|^2\right]}{B}$$

Summing up the above inequality over $j$ proves the result. $\qquad\square$

### F.4 Proof of Step 2

*Proof.* Define $(r_0^i)^* = \nabla_\lambda F(\lambda_H(\theta_0^i))$, for the ease of notation. Then

$$\mathbb{E}\left[\left\|g_0^i - \left[\nabla_\theta \lambda_H(\theta_0^i)\right]^\top r_0^i\right\|^2\right] \tag{31}$$

$$= \mathbb{E}\left[\left\|\frac{1}{N}\sum_{\tau\in\mathcal{N}_i} \widehat{g}(\tau|\theta_0^i, r_0^i) - \frac{1}{N}\sum_{\tau\in\mathcal{N}_i} \widehat{g}(\tau|\theta_0^i, (r_0^i)^*) + \frac{1}{N}\sum_{\tau\in\mathcal{N}_i} \widehat{g}(\tau|\theta_0^i, (r_0^i)^*) - \nabla_\theta F(\lambda_H(\theta_0^i))\right.\right.$$

$$\left.\left. + \nabla_\theta F(\lambda_H(\theta_0^i)) - \left[\nabla_\theta \lambda_H(\theta_0^i)\right]^\top r_0^i\right\|^2\right]$$

$$\leq 3\mathbb{E}\left[\left\|\frac{1}{N}\sum_{\tau\in\mathcal{N}_i}\left(\widehat{g}(\tau|\theta_0^i, r_0^i) - \widehat{g}(\tau|\theta_0^i, (r_0^i)^*)\right)\right\|^2\right] + 3\mathbb{E}\left[\left\|\nabla_\theta F(\lambda_H(\theta_0^i)) - \left[\nabla_\theta \lambda_H(\theta_0^i)\right]^\top r_0^i\right\|^2\right]$$

$$+ 3\mathbb{E}\left[\left\|\frac{1}{N}\sum_{\tau\in\mathcal{N}_i}\widehat{g}(\tau|\theta_0^i, (r_0^i)^*) - \nabla_\theta F(\lambda_H(\theta_0^i))\right\|^2\right].$$

For the first term of (31), we have

$$\mathbb{E}\left[\left\|\frac{1}{N}\sum_{\tau\in\mathcal{N}_i}\left(\widehat{g}(\tau|\theta_0^i, r_0^i) - \widehat{g}(\tau|\theta_0^i, (r_0^i)^*)\right)\right\|^2\right]$$

$$\leq \frac{1}{N}\sum_{\tau\in\mathcal{N}_i}\mathbb{E}\left[\left\|\widehat{g}(\tau|\theta_0^i, r_0^i) - \widehat{g}(\tau|\theta_0^i, (r_0^i)^*)\right\|^2\right]$$

$$\overset{(a)}{\leq} \frac{4\ell_\psi^2}{(1-\gamma)^4} \cdot \mathbb{E}\left[\|r_0^i - (r_0^i)^*\|_\infty^2\right]$$

$$\overset{(b)}{\leq} \frac{4\ell_\psi^2 \cdot L_\lambda^2}{(1-\gamma)^4} \cdot \mathbb{E}\left[\|\lambda_0^i - \lambda_H(\theta_0^i)\|^2\right]$$

$$\overset{(c)}{\leq} \frac{4\ell_\psi^2 \cdot L_\lambda^2}{N(1-\gamma)^6}$$

where (a) is due to Lemma E.2; (b) is because $r_0^i = \nabla_\lambda F(\lambda_0^i)$, $(r_0^i)^* = \nabla_\lambda F(\lambda_H(\theta_0^i))$ and Assumption 5.2; (c) is because Lemma F.1. Similarly, we can show that

$$\mathbb{E}\left[\left\|\frac{1}{N}\sum_{\tau \in \mathcal{N}_i}\widehat{g}(\tau|\theta_0^i,(r_0^i)^*) - \nabla_\theta F(\lambda_H(\theta_0^i))\right\|^2\right] \leq \frac{\ell_{\lambda,\infty}^2}{N(1-\gamma)^4}$$

and

$$\mathbb{E}\left[\left\|\nabla_\theta F(\lambda_H(\theta_0^i)) - \left[\nabla_\theta \lambda_H(\theta_0^i)\right]^\top r_0^i\right\|^2\right] \leq \frac{4\ell_\psi^2 \cdot L_\lambda^2}{N(1-\gamma)^6}.$$

Substituting the above three bounds into (31) proves the inequality (25). □

## G   Proof of Theorem 5.9

*Proof.* Summing up the ascent inequality of Lemma 5.5 for the $i$-th epoch and taking the expectation on both sides, we have

$$\frac{\eta}{4}\sum_{j=0}^{m-1}\mathbb{E}\left[\|\mathcal{G}_\eta(\theta_j^i)\|^2\right]$$

$$\overset{(i)}{\leq} \mathbb{E}\left[F(\lambda(\theta_m^i))\right] - \mathbb{E}\left[F(\lambda(\theta_0^i))\right] - \left(\frac{1}{2\eta} - L_\theta\right) \cdot \sum_{j=0}^{m-1}\mathbb{E}\left[\|\theta_{j+1}^i - \theta_j^i\|^2\right]$$

$$+ \left(\frac{\eta}{2} + \frac{1}{2L_\theta}\right) \cdot \sum_{j=0}^{m-1}\mathbb{E}\left[\|\nabla_\theta F(\lambda(\theta_j^i)) - g_j^i\|^2\right]$$

$$\overset{(ii)}{\leq} \mathbb{E}\left[F(\lambda(\theta_m^i))\right] - \mathbb{E}\left[F(\lambda(\theta_0^i))\right] + m\left(\frac{\eta}{2} + \frac{1}{2L_\theta}\right)\left(\frac{C_1}{N} + \gamma^{2H}C_2\right)$$

$$- \left(\left(\frac{1}{2\eta} - L_\theta\right) - \left(\frac{\eta}{2} + \frac{1}{2L_\theta}\right) \cdot \left(\frac{m}{B}C_3 + C_4\right)\right) \cdot \sum_{j=0}^{m-1}\mathbb{E}\left[\|\nabla_\theta F(\lambda(\theta_j^i)) - g_j^i\|^2\right],$$

where (i) is because Lemma 5.5 and (ii) is because Lemma 5.8. Note that $N = B^2 = m^2$, and $\eta = \frac{1}{1+(C_3+C_4)/L_\theta^2} \cdot \frac{1}{2L_\theta}$, the coefficient

$$\left(\frac{1}{2\eta} - L_\theta\right) - \left(\frac{\eta}{2} + \frac{1}{2L_\theta}\right) \cdot \left(\frac{m}{B}C_3 + C_4\right) \geq 0.$$

Hence we have

$$\frac{\eta}{4}\sum_{j=0}^{m-1}\mathbb{E}\left[\|\mathcal{G}_\eta(\theta_j^i)\|^2\right] \leq \mathbb{E}\left[F(\lambda(\theta_m^i))\right] - \mathbb{E}\left[F(\lambda(\theta_0^i))\right] + m\left(\frac{\eta}{2} + \frac{1}{2L_\theta}\right)\left(\frac{C_1}{N} + \gamma^{2H}C_2\right).$$

Note that $\lambda(\theta_0^i) = \tilde{\theta}_{i-1}$ and $\lambda(\theta_m^i) = \tilde{\theta}_i$, summing the above inequality over all $T$ epochs and dividing both sides with $\frac{\eta}{4} \cdot Tm$ yields

$$\frac{1}{Tm}\sum_{j=0}^{m-1}\sum_{i=1}^{T}\mathbb{E}\left[\|\mathcal{G}_\eta(\theta_j^i)\|^2\right] \leq \frac{4(F(\lambda(\theta^*)) - F(\lambda(\theta_0^i)))}{Tm \cdot \eta} + \left(2 + \frac{2}{L_\theta \eta}\right)\left(\frac{C_1}{N} + \gamma^{2H}C_2\right).$$

Choosing $T = m = \epsilon^{-1}$, $H = \frac{2\log(\epsilon^{-1})}{1-\gamma}$ and let $\theta_{out}$ be selected uniformly at random from $\{\theta_j^i : i = 1, ..., T, j = 0, ..., m-1\}$ yields

$$\mathbb{E}\left[\|\mathcal{G}_\eta(\theta_{out})\|^2\right]$$

$$= \frac{1}{Tm}\sum_{j=0}^{m-1}\sum_{i=1}^{T}\mathbb{E}\left[\|\mathcal{G}_\eta(\theta_j^i)\|^2\right]$$

$$\leq \frac{4(F(\lambda(\theta^*)) - F(\lambda(\theta_0^i)))}{Tm \cdot \eta} + \left(2 + \frac{2}{L_\theta \eta}\right)\left(\frac{C_1}{N} + \gamma^{2H}C_2\right)$$

$$= \left(4\eta^{-1} \cdot (F(\lambda(\theta^*)) - F(\lambda(\theta_0^i))) + (6 + (C_3 + C_4)/L_\theta^2) \cdot (C_3 + \gamma^{2H}\epsilon^{-2}C_4)\right) \cdot \epsilon^2$$

$$= \mathcal{O}(\epsilon^2).$$

Then Lemma 5.4 indicates that $\mathbb{E}\big[\|\nabla_\theta F(\lambda(\theta_{out}))\|^2\big] \le \mathcal{O}(\epsilon^2)$. Then by Jensen's inequality,

$$\mathbb{E}\big[\|\nabla_\theta F(\lambda(\theta_{out}))\|\big] \le \sqrt{\mathbb{E}\big[\|\nabla_\theta F(\lambda(\theta_{out}))\|^2\big]} \le \mathcal{O}(\epsilon).$$

Hence we complete the proof. $\qquad\square$

## H  Proof of Lemma 5.12

*Proof.* By Lemma 5.3, the function $F \circ \lambda(\cdot)$ is $L_\theta$-smooth. Then we have

$$
\begin{aligned}
& |F(\lambda(\theta)) - F(\lambda(\theta_j^i)) - \langle g_j^i, \theta - \theta_j^i \rangle| \\
\le\ & |F(\lambda(\theta)) - F(\lambda(\theta_j^i)) - \langle \nabla_\theta F(\lambda(\theta_j^i)), \theta - \theta_j^i \rangle| + |\langle \nabla_\theta F(\lambda(\theta_j^i)) - g_j^i, \theta - \theta_j^i \rangle| \\
\le\ & \frac{L_\theta}{2}\|\theta - \theta_j^i\|^2 + \frac{L_\theta}{2}\|\theta - \theta_j^i\|^2 + \frac{1}{2L_\theta}\|\nabla_\theta F(\lambda(\theta_j^i)) - g_j^i\|^2 \\
=\ & L_\theta\|\theta - \theta_j^i\|^2 + \frac{1}{2L_\theta}\|\nabla_\theta F(\lambda(\theta_j^i)) - g_j^i\|^2.
\end{aligned}
$$

That is,

$$
\begin{aligned}
& F(\lambda(\theta_j^i)) + \langle g_j^i, \theta - \theta_j^i \rangle + L_\theta\|\theta - \theta_j^i\|^2 + \frac{1}{2L_\theta}\|\nabla_\theta F(\lambda(\theta_j^i)) - g_j^i\|^2 \\
\ge\ & F(\lambda(\theta)) \\
\ge\ & F(\lambda(\theta_j^i)) + \langle g_j^i, \theta - \theta_j^i \rangle - L_\theta\|\theta - \theta_j^i\|^2 - \frac{1}{2L_\theta}\|\nabla_\theta F(\lambda(\theta_j^i)) - g_j^i\|^2.
\end{aligned}
\tag{32}
$$

By the discussion of (12), our truncated gradient update is equivalent to solving

$$
\theta_{j+1}^i := \operatorname*{argmax}_{\|\theta - \theta_j^i\| \le \delta} F(\lambda(\theta_j^i)) + \langle g_j^i, \theta - \theta_j^i \rangle - \frac{1}{2\eta}\|\theta - \theta_j^i\|^2.
\tag{33}
$$

By (32) and (33), we have

$$
\begin{aligned}
& F(\lambda(\theta_{j+1}^i)) \\
\overset{(a)}{\ge}\ & F(\lambda(\theta_j^i)) + \langle g_j^i, \theta_{j+1}^i - \theta_j^i \rangle - L_\theta\|\theta_{j+1}^i - \theta_j^i\|^2 - \frac{1}{2L_\theta}\|\nabla_\theta F(\lambda(\theta_j^i)) - g_j^i\|^2 \\
=\ & F(\lambda(\theta_j^i)) + \langle g_j^i, \theta_{j+1}^i - \theta_j^i \rangle - \frac{1}{2\eta}\|\theta_{j+1}^i - \theta_j^i\|^2 - \frac{1}{2L_\theta}\|\nabla_\theta F(\lambda(\theta_j^i)) - g_j^i\|^2 \\
& + \Big(\frac{1}{2\eta} - L_\theta\Big) \cdot \|\theta_{j+1}^i - \theta_j^i\|^2 \\
\overset{(b)}{=}\ & \max_{\|\theta - \theta_j^i\| \le \delta}\Big\{ F(\lambda(\theta_j^i)) + \langle g_j^i, \theta - \theta_j^i \rangle - \frac{1}{2\eta}\|\theta - \theta_j^i\|^2 \Big\} - \frac{1}{2L_\theta}\|\nabla_\theta F(\lambda(\theta_j^i)) - g_j^i\|^2 \\
& + \Big(\frac{1}{2\eta} - L_\theta\Big) \cdot \|\theta_{j+1}^i - \theta_j^i\|^2 \\
\overset{(c)}{\ge}\ & \max_{\|\theta - \theta_j^i\| \le \delta}\Big\{ F(\lambda(\theta)) - \Big(\frac{1}{2\eta} + L_\theta\Big)\|\theta - \theta_j^i\|^2 \Big\} - \frac{1}{L_\theta}\|\nabla_\theta F(\lambda(\theta_j^i)) - g_j^i\|^2 \\
& + \Big(\frac{1}{2\eta} - L_\theta\Big) \cdot \|\theta_{j+1}^i - \theta_j^i\|^2,
\end{aligned}
$$

where (a) is by setting $\theta = \theta_{j+1}^i$ in second half of (32), (b) is due to (33), and (c) is due to the first half of (32). That is,

$$
\begin{aligned}
F(\lambda(\theta_{j+1}^i)) \ge\ & \max_{\|\theta - \theta_j^i\| \le \delta}\Big\{ F(\lambda(\theta)) - \Big(L_\theta + \frac{1}{2\eta}\Big) \cdot \|\theta - \theta_j^i\|^2 \Big\} \\
& + \Big(\frac{1}{2\eta} - L_\theta\Big) \cdot \|\theta_{j+1}^i - \theta_j^i\|^2 - \frac{1}{L_\theta}\|g_j^i - \nabla_\theta F(\lambda(\theta_j^i))\|^2.
\end{aligned}
\tag{34}
$$

For any $\epsilon < \bar{\epsilon}$, by Assumption 5.11, $(1-\epsilon)\lambda(\theta_j^i) + \epsilon\lambda(\theta^*) \in \mathcal{V}_{\lambda(\theta_j^i)}$ and hence

$$\theta_\epsilon := \left(\lambda|_{\mathcal{U}_{\theta_j^i}}\right)^{-1}\left((1-\epsilon)\lambda(\theta_j^i) + \epsilon\lambda(\theta^*)\right) \in \mathcal{U}_{\theta_j^i} \subset B(\theta_j^i, \delta).$$

Consequently, substituting the above $\theta_\epsilon$ into (34) yields

$$\begin{aligned}
F(\lambda(\theta_{j+1}^i)) &\geq F \circ \lambda \circ \left(\lambda|_{\mathcal{U}_{\theta_j^i}}\right)^{-1}\left((1-\epsilon)\lambda(\theta_j^i) + \epsilon\lambda(\theta^*)\right) &(35)\\
&\quad - \left(L_\theta + \frac{1}{2\eta}\right) \cdot \|\psi\left((1-\epsilon) \cdot \lambda(\theta_j^i) + \epsilon \cdot \lambda^*\right) - \theta_j^i\|^2 \\
&\quad + \left(\frac{1}{2\eta} - L_\theta\right) \cdot \|\theta_{j+1}^i - \theta_j^i\|^2 - \frac{1}{L_\theta}\|g_j^i - \nabla_\theta F(\lambda(\theta_j^i))\|^2.
\end{aligned}$$

Note that $\lambda \circ \left(\lambda|_{\mathcal{U}_{\theta_j^i}}\right)^{-1}$ is the identity mapping on $\mathcal{U}_{\theta_j^i}$, then

$$\begin{aligned}
F \circ \lambda \circ \left(\lambda|_{\mathcal{U}_{\theta_j^i}}\right)^{-1}\left((1-\epsilon)\lambda(\theta_j^i) + \epsilon\lambda(\theta^*)\right) &= F\left((1-\epsilon) \cdot \lambda(\theta_j^i) + \epsilon \cdot \lambda(\theta^*)\right) \\
&\geq (1-\epsilon) \cdot F(\lambda(\theta_j^i)) + \epsilon \cdot F(\lambda(\theta^*))
\end{aligned}$$

and

$$\begin{aligned}
&\left\|\left(\lambda|_{\mathcal{U}_{\theta_j^i}}\right)^{-1}\left((1-\epsilon)\lambda(\theta_j^i) + \epsilon\lambda(\theta^*)\right) - \theta_j^i\right\|^2 \\
&= \left\|\left(\lambda|_{\mathcal{U}_{\theta_j^i}}\right)^{-1}\left((1-\epsilon)\lambda(\theta_j^i) + \epsilon\lambda(\theta^*)\right) - \left(\lambda|_{\mathcal{U}_{\theta_j^i}}\right)^{-1}(\lambda(\theta_j^i))\right\|^2 \\
&\overset{(a)}{\leq} \epsilon^2\ell_\theta^2 \cdot \|\lambda(\theta_j^i) - \lambda^*\|^2 \\
&= \epsilon^2\ell_\theta^2 \cdot \left(\|\lambda(\theta_j^i)\|^2 + \|\lambda^*\|^2 - 2\langle\lambda(\theta_j^i), \lambda^*\rangle\right) \\
&\leq \frac{2\epsilon^2\ell_\theta^2}{(1-\gamma)^2}
\end{aligned}$$

where (a) is due to Assumption 5.11. Substituting the above two inequalities into (35) and slightly rearranging the terms yields

$$\begin{aligned}
F(\lambda(\theta^*)) - F(\lambda(\theta_{j+1}^i)) &\leq (1-\epsilon)\left(F(\lambda(\theta^*)) - F(\lambda(\theta_j^i))\right) \\
&\quad + \left(L_\theta + \frac{1}{2\eta}\right)\frac{2\epsilon^2\ell_\theta^2}{(1-\gamma)^2} - \left(\frac{1}{2\eta} - L_\theta\right)\|\theta_{j+1}^i - \theta_j^i\|^2 + \frac{1}{L_\theta}\|g_j^i - \nabla_\theta F(\lambda(\theta_j^i))\|^2,
\end{aligned}$$

which completes the proof. $\qquad\square$

# I   Proof of Theorem 5.13

*Proof.* Recall Lemma 5.12, where we have

$$\begin{aligned}
F(\lambda(\theta^*)) - F(\lambda(\theta_{j+1}^i)) &\leq (1-\epsilon)\left(F(\lambda(\theta^*)) - F(\lambda(\theta_j^i))\right) + \left(L_\theta + \frac{1}{2\eta}\right)\frac{2\epsilon^2\ell_\theta^2}{(1-\gamma)^2} \\
&\quad - \left(\frac{1}{2\eta} - L_\theta\right)\|\theta_{j+1}^i - \theta_j^i\|^2 + \frac{1}{L_\theta}\|g_j^i - \nabla_\theta F(\lambda(\theta_j^i))\|^2 &(36)
\end{aligned}$$

For the ease of notation, let us denote $\sigma_j^i = -\left(\frac{1}{2\eta} - L_\theta\right) \cdot \|\theta_{j+1}^i - \theta_j^i\|^2 + \frac{1}{L_\theta}\|g_j^i - \nabla_\theta F(\lambda(\theta_j^i))\|^2$. Taking the expectation and telescoping (36) over $j$ yields

$$\begin{aligned}
\mathbb{E}\left[F(\lambda(\theta^*)) - F(\lambda(\theta_m^i))\right] &\leq (1-\epsilon)^m \cdot \mathbb{E}\left[F(\lambda(\theta^*)) - F(\lambda(\theta_0^i))\right] + \frac{(2L_\theta + \frac{1}{\eta})\ell_\theta^2}{(1-\gamma)^2} \cdot \epsilon &(37)\\
&\quad + \sum_{j=0}^{m-1}(1-\epsilon)^{m-j-1}\mathbb{E}[\sigma_j^i]
\end{aligned}$$

Next, we show that when the step size $\eta$ is properly chosen, the term $\sum_{j=0}^{m-1}(1-\epsilon)^{m-j-1}\mathbb{E}[\sigma_j^i]$ will be negligible.

$$\sum_{j=0}^{m-1}(1-\epsilon)^{m-j-1}\mathbb{E}[\sigma_j^i]$$

$$=\quad \sum_{j=0}^{m-1}(1-\epsilon)^{m-j-1}\cdot \mathbb{E}\left[-\left(\frac{1}{2\eta}-L_\theta\right)\cdot\|\theta_{j+1}^i-\theta_j^i\|^2 + \frac{1}{L_\theta}\|g_j^i-\nabla_\theta F(\lambda(\theta_j^i))\|^2\right]$$

$$\leq\quad \frac{1}{L_\theta}\sum_{j=0}^{m-1}\mathbb{E}\left[\|g_j^i-\nabla_\theta F(\lambda(\theta_j^i))\|^2\right]-(1-\epsilon)^m\left(\frac{1}{2\eta}-L_\theta\right)\cdot\sum_{j=0}^{m-1}\mathbb{E}\left[\|\theta_{j+1}^i-\theta_j^i\|^2\right]$$

Let us choose $m=\epsilon^{-1}$. Then as long as $\epsilon\leq 1/2$, we have $(1-\epsilon)^{\epsilon^{-1}}\geq\frac{1}{4}$. Then, using Lemma 5.8 yields

$$\sum_{j=0}^{m-1}(1-\epsilon)^{m-j-1}\mathbb{E}[\sigma_j^i]$$

$$\leq\quad \frac{1}{L_\theta}\sum_{j=0}^{m-1}\mathbb{E}\left[\|g_j^i-\nabla_\theta F(\lambda(\theta_j^i))\|^2\right]-\left(\frac{1}{8\eta}-\frac{L_\theta}{4}\right)\cdot\sum_{j=0}^{m-1}\mathbb{E}\left[\|\theta_{j+1}^i-\theta_j^i\|^2\right]$$

$$\leq\quad \frac{mC_1}{L_\theta N}+\frac{mC_2\cdot\gamma^{2H}}{L_\theta}+\frac{C_3}{L_\theta B}\cdot\sum_{j=0}^{m-1}\sum_{j'=0}^{j-1}\mathbb{E}\left[\|\theta_{j'+1}^i-\theta_{j'}^i\|^2\right]-\left(\frac{1}{8\eta}-\frac{L_\theta}{4}-\frac{C_4}{L_\theta}\right)\cdot\sum_{j=0}^{m-1}\mathbb{E}\left[\|\theta_{j+1}^i-\theta_j^i\|^2\right]$$

$$\leq\quad \frac{mC_1}{L_\theta N}+\frac{mC_2\cdot\gamma^{2H}}{L_\theta}-\left(\frac{1}{8\eta}-\frac{L_\theta}{4}-\frac{mC_3}{L_\theta B}-\frac{C_4}{L_\theta}\right)\cdot\sum_{j=0}^{m-1}\mathbb{E}\left[\|\theta_{j+1}^i-\theta_j^i\|^2\right]$$

Since we choose $\eta\leq\frac{1}{2L_\theta+\frac{8(C_3+C_4)}{L_\theta}}$, then $\frac{1}{8\eta}-\frac{L_\theta}{4}-\frac{mC_3}{L_\theta B}-\frac{C_4}{L_\theta}$. Because we choose $B=m=\epsilon^{-1}$ and $N=\epsilon^{-2}$, then

$$\sum_{j=0}^{m-1}(1-\epsilon)^{m-j-1}\mathbb{E}[\sigma_j^i]\leq\frac{C_1}{L_\theta}\epsilon+\frac{C_2}{L_\theta\epsilon}\cdot\gamma^{2H}.$$

Substituting the above inequality into (37) and use the fact that $(1-\epsilon)^{\epsilon^{-1}}\leq\frac{1}{2},\forall\epsilon\leq 1$ yields

$$\mathbb{E}\left[F(\lambda(\theta^*))-F(\lambda(\theta_m^i))\right]\leq\frac{1}{2}\mathbb{E}\left[F(\lambda(\theta^*))-F(\lambda(\theta_0^i))\right]+\left(\frac{(2L_\theta+\frac{1}{\eta})\ell_\theta^2}{(1-\gamma)^2}+\frac{C_1}{L_\theta}\right)\cdot\epsilon+\frac{C_2}{L_\theta\epsilon}\cdot\gamma^{2H}.$$

Using the fact that $\tilde{\theta}_{i-1}=\theta_0^i$ and $\tilde{\theta}_i=\theta_m^i$ proves

$$\mathbb{E}\left[F(\lambda(\theta^*))-F(\lambda(\tilde{\theta}_i))\right]\leq\frac{1}{2}\mathbb{E}\left[F(\lambda(\theta^*))-F(\lambda(\tilde{\theta}_{i-1}))\right]+\left(\frac{(2L_\theta+\frac{1}{\eta})\ell_\theta^2}{(1-\gamma)^2}+\frac{C_1}{L_\theta}\right)\cdot\epsilon+\frac{C_2}{L_\theta\epsilon}\cdot\gamma^{2H}.$$

Again, telescoping sum the above inequality for $i=1,...,T$ proves

$$\mathbb{E}\left[F(\lambda(\theta^*))-F(\lambda(\tilde{\theta}_T))\right]\leq\left(F(\lambda(\theta^*))-F(\lambda(\tilde{\theta}_0))+\frac{(4L_\theta+\frac{2}{\eta})\ell_\theta^2}{(1-\gamma)^2}+\frac{2C_1}{L_\theta}\right)\cdot\frac{1}{2^T}+\frac{2C_2}{L_\theta\epsilon}\cdot\gamma^{2H},$$

Note that we choose $T=\log_2(\epsilon^{-1})$ and $H=\frac{2\log(1/\epsilon)}{1-\gamma}$, indicating that $\frac{1}{2^T}=\epsilon$ and $\gamma^{2H}=\mathcal{O}(\epsilon^4)$. Therefore, $\mathbb{E}\left[F(\lambda(\theta^*))-F(\lambda(\tilde{\theta}_T))\right]\leq\mathcal{O}(\epsilon)$. $\qquad\square$

## J  Experiment Settings

### J.1  Experiment Setting for Ordinary RL Tasks

The snapshot batchsize $N$ is chosen from the grid search from $\{10,25,50,100,200\}$. $B$ and $m$ are calculated according to the provided formula of the theory. The learning rate $\eta$ is also chosen from a grid search from the range $[10^{-4},10^{-1}]$. More details are presented in the list below.

| Environment | Algorithm | Policy Network | $\gamma$ | $H$ | $B$ | $N$ | $m$ | $\eta$ | $\delta$ |
|---|---|---|---|---|---|---|---|---|---|
| CartPole-v0 | TSIVR-PG | | | | 5 | 25 | 5 | $1 \times 10^{-2}$ | $1 \times 10^{-2}$ |
| | REINFORCE | | | | | 25 | | $5 \times 10^{-3}$ | |
| | SVRPG | $4 \times 64 \times 64 \times 2$ | 0.99 | 200 | 8 | 25 | 3 | $5 \times 10^{-3}$ | |
| | SRVR-PG | | | | 5 | 25 | 5 | $5 \times 10^{-3}$ | |
| | HSPGA | | | | 5 | 25 | 5 | $8 \times 10^{-3}$ | |
| FrozenLake8x8 | TSIVR-PG | | | | 10 | 100 | 10 | $1 \times 10^{-1}$ | $1 \times 10^{-2}$ |
| | REINFORCE | | | | | 100 | | $5 \times 10^{-2}$ | |
| | SVRPG | $64 \times 4$ | 0.99 | 200 | 20 | 100 | 5 | $5 \times 10^{-2}$ | |
| | SRVR-PG | | | | 10 | 100 | 10 | $5 \times 10^{-2}$ | |
| | HSPGA | | | | 10 | 100 | 10 | $8 \times 10^{-2}$ | |
| Acrobot-v1 | TSIVR-PG | | | | 10 | 100 | 10 | $2 \times 10^{-3}$ | $1 \times 10^{-2}$ |
| | REINFORCE | | | | | 100 | | $2 \times 10^{-3}$ | |
| | SVRPG | $4 \times 64 \times 64 \times 3$ | 0.999 | 500 | 20 | 100 | 5 | $2 \times 10^{-3}$ | |
| | SRVR-PG | | | | 10 | 100 | 10 | $2 \times 10^{-3}$ | |
| | HSPGA | | | | 10 | 100 | 10 | $2 \times 10^{-3}$ | |

## J.2   Experiment Setting for Maximizing Non-linear Objective Function

The parameter selection is performed in the same way as the previous section. The inner planning loop of MaxEnt is performed by a single policy gradient step. The details are presented in the list below.

| Algorithm | Policy Network | $\gamma$ | $H$ | $B$ | $N$ | $m$ | $\eta$ | $\delta$ |
|---|---|---|---|---|---|---|---|---|
| TSIVR-PG | $64 \times 4$ | 0.99 | 200 | 20 | 100 | 5 | $5 \times 10^{-2}$ | $9 \times 10^{-2}$ |
| MaxEnt | $64 \times 4$ | 0.99 | 200 | | 100 | | $9 \times 10^{-2}$ | |