# OpenReview forum: "On the Convergence and Sample Efficiency of Variance-Reduced Policy Gradient Method"
_NeurIPS.cc/2021/Conference — NeurIPS 2021 Spotlight_

### Official Review · Reviewer_vT8s · 2021-07-07

**Rating:** 8
**Confidence:** 4

**Summary:**

This paper investigates a novel truncation mechanism to perform variance reduction in policy gradient methods without the uncheckable importance weight assumption widely used in the literature. This technique is analyzed in depth in theory and supported with strong numerical experiments.

**Limitations And Societal Impact:**

The authors adequately addressed the limitations and potential negative societal impact of their work.

**Main Review:**

First, I would like to thank the authors for their work: It is a very-well written and clear paper with relevant motivations (clear introduction), well-explained related work and fine overall structure. I really enjoyed reading this paper.

- Originality:
The idea behind TSIVR-PG is originial and there are solid contributions. This paper presents a new truncation mechanism for policy gradient methods which extends several convergence results known so far in restricted settings. The related work is adequately cited and clearly describes how the current paper differs from it.

- Quality:
Technically strong, highly general results, advanced techniques. The different examples of the problem formulation are convincing. The numerical experiments strongly support the theoretical claims.

- Clarity:
The paper is very well written with fine overall structure making it easy for the reader to grasp all the main ideas.
The mathematical proofs are well polished.

- Significance:
The results are quite important as they extend and relax several assumptions made so far in policy gradient methods.

- Typos:
abstract "overparameterizaiton"
l.113 tuncation mechanism [is] proposed
l.293 beyongd

**Time Spent Reviewing:**

4

---

> ### Author Response · Authors · 2021-08-04
> **Response to Reviewer vT8s**
>
> We thank the reviewer for appreciating our results. We will correct all these typos pointed out by the reviewer. We will also carefully go through the paper again in order to correct all other typos.

---

### Official Review · Reviewer_sR2P · 2021-07-12

**Rating:** 6
**Confidence:** 4

**Summary:**

This paper proposed an algorithm TSIVR-PG, which can provably control the importance weights, while other methods rely on uncheckable ones when it comes to variance reduction in off-policy estimation. To support the idea, derivations are given in Section 5 to theoretically prove the convergence of this model, and experiments in Section 6 also support its effectiveness.

**Limitations And Societal Impact:**

Though this method is applicable to discrete space small-sized problems, it is still not clear why the continuous space problems and neural networks are not taken into accounts. Besides, there are several points that are not explained clearly and may cause limitations on understanding.
1. In Eq. 10, why the update of $g_j$ has to use $r_{j-1}$, what if $r_{j}$ is used in Eq. 10 given that $r_j$ can be updated/obtained according to Eq. 9 ？
2.  The total numbers of samples used in Theorem 5.9 and 5.13 are obtained according to algorithm 1.  I can follow Theorem 5.13 but get confused about Theorem 5.9. Could you please explain more about how the total number of samples is obtained in Theorem 5.9?


**Main Review:**

This work incorporates a simple gradient truncation mechanism into importance weights control for variance reducing, which addresses the restrictive assumption issues that other methods suffer. The mathematical derivation of convergence seems solid. The work seems good to me, however, there could be several points that could further improve the overall quality of this paper, as follows:
1. Some symbols are used without an explicit statement when first mentioned in this paper,  e.g.,  $\gamma$ and $\mathbb{P}$ in equation 3.
2. In the denominator part of equation 5, the position of $h$ and $t$ is supposed to be exchanged.
3. Above line 169, a redundant $\theta$ is added in the definitive equation of $\theta_{+}$.
4. In equation 13, the character $L_{h}$ is supposed to be $L_{\psi}$.
5. Inconsistent format, e.g., the figures in Section 6 are not referred to in a correct fashion in the main manuscript.	(Labeled as Fig. 1 but referred to as Fig. 6.1)




**Time Spent Reviewing:**

6 hours

---

> ### Author Response · Authors · 2021-08-04
> **Response to Reviewer sR2P**
>
> We thank the reviewer for pointing out the issues in our presentation.
>
> Comment 1. Some symbols are used without an explicit statement when first mentioned in this paper, e.g., $\gamma$ and $\mathbb{P}$ in eq. 3.
>
> Response:   We will add carefully go through the paper to guarantee all symbols are properly explained when they are used. In particular, after eq. 3, we will add the following explanation: "where $\gamma\in(0,1)$ stands for the discount factor and $\mathbb{P}(\cdot)$ denotes the probability."
>
> Comment 2-5: Various typos or notational mistakes.    Response: We will correct all these issues.
>
> Question: In Eq. 10, why the update of $g_j$ has to use $r_{j-1}$, what if $r_j$ is used,...
>
> Response: This is mainly due to the independence requirement in the technical analysis. In line 150, we explained that we use $r_{j-2}$ instead of $r_{j-1}$ for independence issue. The reason why we use $r_{j-1}$ instead of $r_{j}$ is the same. Such independence is required in many steps of analysis, see e.g. Line 586-588 (they will not be 0 if we use the most updated $r$ estimators).
>
> Question: Could you please explain more about how the total number of samples is obtained in Theorem 5.9?
>
> Response: We are sorry that the formula "$Tm(BH+N)$" is a mistake. It should actually take the same form as that of Theorem 5.13. i.e., it should be $T\times((m-1)B+N) \times H = \tilde{\mathcal{O}}(\epsilon^{-3})$, where $T = \epsilon^{-1}$, $m = B = \epsilon^{-1}$, $N = \epsilon^{-2}$ and $H = \mathcal{O}(\log\epsilon^{-1})$
>
> General question: Though this method is applicable to discrete space small-sized problems, it is still not clear why the continuous space problems and neural networks are not taken into accounts.
>
> Response: The reason why continuous space is not considered is actually explained in Section 8 Limitation. The reason is that our algorithm needs to estimate the occupancy measure and then compute the policy gradient of a general utility. However, estimating the occupancy measure cannot be easily done in the continuous space. We consider this as a potential future direction. Regarding the neural network, we actually considered it as a special case, see Line 19-20. In our experiment, we also use neural networks as the parameterization function $\psi$.

---

> > ### Comment · Reviewer_sR2P · 2021-08-23
> > **Update Theorem 5.9 in your manuscript**
> >
> > Thanks for the clarification. Please update Theorem 5.9 in your manuscript.

---

### Official Review · Reviewer_gxo4 · 2021-07-12

**Rating:** 7
**Confidence:** 3

**Summary:**

In this paper, the authors propose a new policy gradient algorithm called TSIVR-PG for the policy optimization for a general utility function. They adopt a gradient truncation mechanism in TSIVR-PG to get rid of the uncheckable importance weights assumption which is frequently used in previous literatures. They establish convergence rate to stationary points, and global convergence results with additional assumptions. Moreover, they conduct experiments and compare with other previous methods.

**Limitations And Societal Impact:**

I think the authors adequately addressed the limitations and potential negative societal impact of their work.

**Main Review:**

This paper is well-written, highlights their contribution clearly, and has a detailed discussion on related works. I’m glad to see that convergence of the algorithm does not rely on the importance weight assumption that is indeed problematic sometimes. I think the contribution on the theory side is significant. Besides, the empirical evaluation is promising and makes this paper more solid.

Concerns:

(1) I’m wondering if Assump. 5.11 is still too strong? Are there any policy function classes examples that can (approximately) satisfy that assumption?

(2) I'm kind of worried about whether the choice of B and m is fair enough without grid search because the constant in the $O(\cdot)$ also matters and making sure the alignment of $\epsilon$ may not be enough, especially B and m are small in these experiments. I wonder if the authors have some remarks on that?

But considering that the main contribution of this paper is on the theory side, this is not an important issue.


Minor Issues:
Line 168: a redundant \theta after “\theta^+=”

**Time Spent Reviewing:**

8

---

> ### Author Response · Authors · 2021-08-04
> **Response to Reviewer gxo4**
>
> We thank the reviewer for her/his insightful review.
>
> Comment:  I’m wondering if Assump. 5.11 is still too strong? Are there any policy function classes examples that can (approximately) satisfy that assumption?
>
> Response: Although Assumption 5.11 is weaker than the bijection assumption of Variational policy gradient method for reinforcement learning with general utilities (Zhang et al.) (referred to as VPG paper), we do admit that this assumption is still strong to some degree. Currently, it is known that the direct parameterization in tabular MDP satisfies the bijection assumption (Assumption 1 & Appendix H of the VPG paper), and hence it satisfies Assumption 5.11. When it comes to the soft-max type parameterization, this becomes tricky. For example, in tabular soft-max case where $\psi(s,a;\theta) = \theta_{s,a}$, one can verify that a continuous local inverse function can be defined, yet the Lipschitz constant $\ell_\theta$ is hard to compute.
>
> Comment: I'm kind of worried about whether the choice of $B$ and $m$ is fair enough without grid search because the constant in the $O(\cdot)$ also matters and making sure the alignment of $\epsilon$ may not be enough, especially $B$ and $m$ are small in these experiments. I wonder if the authors have some remarks on that?
>
> Response:  We thank the reviewer for pointing this issue out. In the current theorem, we mainly focus on deriving the complexity in terms of $\epsilon$ while not paying attention to the other constants. Currently, we simply set $B = m = \epsilon^{-1}$ and $N = \epsilon^{-2}$. However, if we set $B = c_1\epsilon^{-1}$, $m = c_2\epsilon^{-1}$, and $N = c_3\epsilon^{-1}$ for proper constants $c_1,c_2,c_3$, the constants dependence hidden in the notation $\tilde{\mathcal{O}}(\cdot)$ can indeed be further improved. We will make an additional remark in the revision on this point.
>
> In terms of the grid search in the experiments, this is mainly because we want to make a fair comparison between different methods. The batch sizes are small in the experiment because these problems may not be hard enough, so that small batch sizes work best. In general, one does not need to use the optimal batch sizes for TSIVR-PG to work. See section 6.2, in this experiment, we simply set $B = m = \sqrt{N}$, which corresponds to setting $B = m = \sqrt{N}=\epsilon^{-1}$.  In this case, the algorithm still performs well without grid searching the batch sizes.

---

### Official Review · Reviewer_6pHy · 2021-07-14

**Rating:** 7
**Confidence:** 3

**Summary:**

There is a lot of interest in studying policy gradient type of algorithms for solving the RL problem recently.  This paper develops a variance reduced policy gradient method, where the objective function can be some general utility function instead of just being the cumulative reward. A sample complexity of $\mathcal{O}(\epsilon^{-3})$ was established under some mild conditions, and it was further improved to $\mathcal{O}(\epsilon^{-2})$ under a convexity assumption and a so-called over-parametrization assumption.

**Limitations And Societal Impact:**

No Societal Impact

**Main Review:**

This paper is nicely written. One comment about the structure in Section 5. I understand that the results in Section 5 are presented linearly. However, as a reader, I prefer to first read the main theorem, and then its proof or proof sketch.

When the objective function is the cumulative reward, the sample complexity of policy gradient (or natural policy gradient) was established in the literature, which is $\mathcal{O}(\epsilon^{-2})$ (Lan, G. (2021). Policy mirror descent for reinforcement learning: Linear convergence, new sampling complexity, and generalized problem classes. arXiv preprint arXiv:2102.00135.). However, in this setting, the proposed algorithm in this paper requires the overparametrization assumption to establish the $\mathcal{O}(\epsilon^{-2})$ sample complexity. Is this assumption really necessary or it is an artifact of the proof?

----------------------------------After Feedback----------------------------------

Thank the author for their feedback. I do not have further concerns and I am going to keep my score.

**Time Spent Reviewing:**

2

---

> ### Author Response · Authors · 2021-08-04
> **Response to Reviewer 6pHy**
>
> We thank the reviewer for pointing out the interesting literature of the policy mirror descent (PMD) method.
>
> Comment: As a reader, I prefer to first read the main theorem, and then its proof or proof sketch.
>
> Response: We will rearrange the paper in the revision so that the readers can see the main results first and then its analysis details.
>
> Comment about the PMD method.
>
> Response: In the PMD paper, the author considered the tabular MDP where no parameterization is used. This is also called direct parameterization since we assign a parameter $\pi(a|s)$ for each pair of $(s,a)$. In this case, one can check that the direct parameterization satisfies the overparametrization assumption in our paper. Actually, it even satisfies a stronger bijection assumption, see Assumption 1 and Appendix H of Variational policy gradient method for reinforcement learning with general utilities (Zhang et al.)

---

### Decision · Program_Chairs · 2021-09-27

**Decision:**

Accept (Spotlight)

**Comment:**

The paper introduces a novel truncation mechanism for variance reduction in policy gradient methods. This truncation mechanism allows one to dispense of the importance weight assumption. The paper then develops a complexity theory both for reaching stationary points and global optima (under additional convexity assumptions). All the reviewers agreed that this is an importance contribution, that the theory is sound and well written, and that the additional numerical experiments insightful.